DOI: 10.1038/s41467-018-05910-1　　**OPEN**

# NF-κB inhibition rescues cardiac function by remodeling calcium genes in a Duchenne muscular dystrophy model

Jennifer M. Peterson[1,2,3,7], David J. Wang[1,3,8], Vikram Shettigar[2,3,4], Steve R. Roof[2,3,4,9], Benjamin D. Canan[2,3,4], Nadine Bakkar[1,2,3,10], Jonathan Shintaku[1,2,3,11], Jin-Mo Gu[1,2,3,12], Sean C. Little[2,3,4,13], Nivedita M. Ratnam[1,3], Priya Londhe[1,2,3,14], Leina Lu[5,15], Christopher E. Gaw[3,16], Jennifer M. Petrosino[2,3], Sandya Liyanarachchi[1,3], Huating Wang[5], Paul M.L. Janssen[2,3,4], Jonathan P. Davis[2,3,4], Mark T. Ziolo[2,3,4], Sudarshana M. Sharma[6] & Denis C. Guttridge[1,2,3,8]

Duchenne muscular dystrophy (DMD) is a neuromuscular disorder causing progressive muscle degeneration. Although cardiomyopathy is a leading mortality cause in DMD patients, the mechanisms underlying heart failure are not well understood. Previously, we showed that NF-κB exacerbates DMD skeletal muscle pathology by promoting inflammation and impairing new muscle growth. Here, we show that NF-κB is activated in murine dystrophic (*mdx*) hearts, and that cardiomyocyte ablation of NF-κB rescues cardiac function. This physiological improvement is associated with a signature of upregulated calcium genes, coinciding with global enrichment of permissive H3K27 acetylation chromatin marks and depletion of the transcriptional repressors CCCTC-binding factor, SIN3 transcription regulator family member A, and histone deacetylase 1. In this respect, in DMD hearts, NF-κB acts differently from its established role as a transcriptional activator, instead promoting global changes in the chromatin landscape to regulate calcium genes and cardiac function.

[1] Department of Cancer Biology and Genetics, Columbus, OH 43210, USA. [2] Center for Muscle Health and Neuromuscular Disorders, Columbus, OH 43210, USA. [3] The Ohio State University Medical Center, Columbus, OH 43210, USA. [4] Department of Physiology and Cell Biology, The Ohio State University Medical Center, Columbus 43210 Ohio, USA. [5] Li Ka Shing Institute of Health Sciences, The Chinese University of Hong Kong, Hong Kong, China. [6] Department of Biochemistry and Molecular Biology, Medical University of South Carolina, Charleston, SC 29425, USA. [7] Present address: Department of Pharmacy and Pharmaceutical Sciences, SUNY Binghamton University, Binghamton, NY 13902, USA. [8] Present address: Department of Pediatrics, Medical University of South Carolina, Charleston, South Carolina 29425, USA. [9] Present address: Q Test Labs, Columbus, OH 43235, USA. [10] Present address: Department of Neurobiology, St Joseph's Hospital and Medical Center-Barrow Neurological Institute, Phoenix, AZ 85013, USA. [11] Present address: Department of Neurology, Columbia University Medical Center, New York, NY 10032, USA. [12] Present address: Department of Biomedical Engineering and Pediatrics, Emory University, Decatur, GA 30322, USA. [13] Present address: Bristol-Myers Squibb, Wallingford, CT 06492, USA. [14] Present address: Molecular Oncology Research Institute, Tufts Medical Center, Boston, MA 02111, USA. [15] Present address: Department of Genetics and Genome Sciences, Case Western Reserve University, Cleveland, OH 44106, USA. [16] Present address: Children's Hospital of Philadelphia, Philadelphia, PA 19104, USA. Correspondence and requests for materials should be addressed to D.C.G. (email: guttridg@musc.edu)

Dystrophin is a large cytoplasmic protein that provides structural integrity to muscle. Mutations in the dystrophin gene result in X-linked dilated cardiomyopathy, Becker muscular dystrophy, and Duchenne muscular dystrophy (DMD)[1–3]. Loss of dystrophin protein has also been reported in acquired heart failure, myocardial infection, and myocardial infarction patients, as well as in animal models of myocardial injury and infection[4–8]. This suggests that downstream effects of dystrophin loss is a common link in cardiomyopathy. In DMD patients and animal models, limb, diaphragm, and cardiac muscles are all progressively affected, resulting in loss of ambulation, respiratory distress, and cardiomyopathy, respectively. Cardiac involvement in patients begins in the second to third decade of life and is a leading cause of morbidity and mortality[9]. These patients commonly develop cardiomyopathy with arrhythmias, and animal studies show that calcium dysregulation is a causative factor in the pathology[10,11]. However, the molecular mechanisms underlying disease development in these patients have not been elucidated, and the acute cardiac events that lead to an accumulation of cardiac damage are not well understood.

NF-κB is ubiquitously expressed and functions in cell survival, apoptosis, growth, and differentiation[12]. Our group and others have shown that NF-κB signaling regulates both physiological (differentiation, growth, and metabolism) and pathophysiological (cachexia, atrophy, and dystrophy) aspects of skeletal muscle biology[13–16]. The most common NF-κB complex is the p50/p65 heterodimer, with p65 containing the transactivation domain required to mediate transcriptional activation. NF-κB is held in tight regulation by the inhibitory protein IκB. The upstream IKK kinase (inhibitor of NF-κB kinase) is comprised of two catalytic subunits, IKKα and IKKβ, and a regulatory subunit, IKKγ/NEMO. During classical NF-κB activation, IKK is phosphorylated, and in turn phosphorylates IκB. This phosphorylation causes ubiquitination and subsequent degradation of IκB by the 26S proteasome pathway. Degradation of IκB unmasks a nuclear localization site, allowing NF-κB to translocate to the nucleus, bind consensus sites on target genes, and activate gene expression.

Studies show that classical NF-κB promotes skeletal muscle pathology in the *mdx* murine DMD model[17–21]. Inhibiting NF-κB, either globally by ablating one copy of the p65 gene or conditionally by removing IKKβ alleles in skeletal muscle fibers or myeloid cells, improved dystrophic pathology[18]. Cell or gene therapy designed to target p65 or IKK, respectively, were also effective in protecting *mdx* skeletal muscle[19–21]. Mechanistically, inhibiting NF-κB improved the histology and function of dystrophic limb and diaphragm muscles by enhancing the regenerative potential of muscle stem cells and reducing muscle damage from inflammatory cells[18]. These findings laid the foundation for inhibiting NF-κB as a therapy to treat DMD.

As a therapeutic, we and others have examined the potential of using the Nemo Binding Domain (NBD) peptide, which is a specific NF-κB inhibitor[22]. In *mdx* mice, NBD treatment rescued diaphragm function and improved overall muscle endurance[17,18]. Histologically, NBD reduced inflammation and enhanced regeneration in limb muscles, indicating dystrophic muscles were stabilized. In the Golden Retriever DMD model (GRMD), NBD administration improved hind-limb function, posture, and skeletal muscle histopathology[23]. While encouraged that such results might translate to improved ambulation and respiratory function in DMD patients, we realized that further development of an NF-κB inhibitor would require investigation into its effects on dystrophic cardiac muscle. In a preliminary study, we showed that NBD administration rescued in vitro cardiac function in the severe dystrophin/utrophin double knockout murine DMD model[24]. These results indicated that NF-κB contributes to dystrophic cardiac disease, but how it promotes this pathology remains unknown.

In this study, we set out to understand the mechanism by which NF-κB regulates cardiac dysfunction in *mdx* mice. We discover that unlike the typical function of NF-κB as an inducer of gene expression, NF-κB ablation in *mdx* cardiomyocytes causes global permissive chromatin remodeling on enhancers of calcium genes, which in turn permits increased gene expression and an overall improvement in cardiac function.

## Results

**NF-κB promotes cardiac dysfunction in *mdx* mice.** Cardiomyopathy develops progressively over the lifespan of the *mdx* mouse. From a young age (8–10 weeks), only ex vivo cardiomyocyte dysfunction is detected[25,26]. Histological abnormalities are apparent by 7 months[27], and by 10–12 months, dysfunction in vivo is detectable by echocardiogram[28–30]. By 2 years, severe cardiomyopathy is apparent[31]. Based on these time points, we asked if NF-κB activation occurs early in the disease process. NF-κB DNA binding was low in 3-month-old adult wt hearts, and increased in age matched *mdx* hearts (Fig. 1a). This activity was attributable to p65 and p50 (Fig. 1a). Phosphorylation of p65 on serine 536, which is a marker of NF-κB transcriptional activity, indicated that NF-κB activation persisted in 1-year-old *mdx* hearts (Fig. 1b). To determine the cellular source of NF-κB, we performed immunohistochemistry on cardiac tissue. Results showed that NF-κB was most prominently localized to *mdx* cardiomyocyte nuclei (Fig. 1c and Supplementary Fig. 1A). Together, these results indicate that NF-κB is active prior to the onset of *mdx* cardiomyopathy, and persists throughout the disease process.

Since we detected activated NF-κB within cardiomyocytes, we asked if this signaling was relevant for dystrophic cardiomyopathy by conditionally deleting IKKβ from *mdx* cardiomyocytes. Knock-out mice were denoted *mdx*[HRTΔIKKβ], while littermates with intact NF-κB were designated *mdx*[IKKβf/f]. We confirmed that IKKβ expression was reduced in *mdx*[HRTΔIKKβ] hearts (Supplementary Fig. 1B). Then we examined gross differences between wt, *mdx*[IKKβf/f] and *mdx*[HRTΔIKKβ] mice. Representative histological whole hearts sections are included in Supplementary Fig. 1C. By 12–14 months, body weight and heart weight adjusted for tibia length from both *mdx*[IKKβf/f] and *mdx*[HRTΔIKKβ] mice were less than wt, but this difference only reached significance in the *mdx*[HRTΔIKKβ] mice (Table 1). These data indicate that cardiomyocyte deletion of IKKβ does not prevent the reduction of body or heart weight that occurs in aging *mdx* mice[30,32].

In vivo dysfunction has been shown at 10 months in *mdx* mice[28–30]. We asked if deleting NF-κB from *mdx* cardiomyocytes resulted in long-term functional changes. To assure we could detect differences from control mice, echocardiograms were performed on 13–14 month old mice. While cardiac output was reduced in *mdx*[IKKβf/f] mice, it was restored to wt levels in *mdx*[HRTΔIKKβ] mice (Fig. 1d). Because anesthetized heart rates were similar between groups (Table 1), stroke volume, which was also significantly compromised in *mdx*[IKKβf/f] mice, was restored to wt values in *mdx*[HRTΔIKKβ] mice (Fig. 1e). Ejection fraction was preserved in *mdx*[IKKβf/f] mice (Fig. 1f), suggesting that *mdx* mice suffer from diastolic dysfunction with either preserved systolic function or a compensation that masks systolic dysfunction. A closer analysis of chambers parameters further supported these findings. While systolic diameters and volumes were not significantly altered (Table 1), end diastolic diameters and volumes were both reduced in *mdx*[IKKβf/f] and normalized in *mdx*[HRTΔIKKβ] mice (Fig. 1g,h), indicating that NF-κB deletion prevented the developed pathology of small cardiac chambers in *mdx* mice. Together, data suggest that in *mdx* cardiomyocytes, NF-κB activation causes a smaller, stiff left ventricle with decreased compliance.

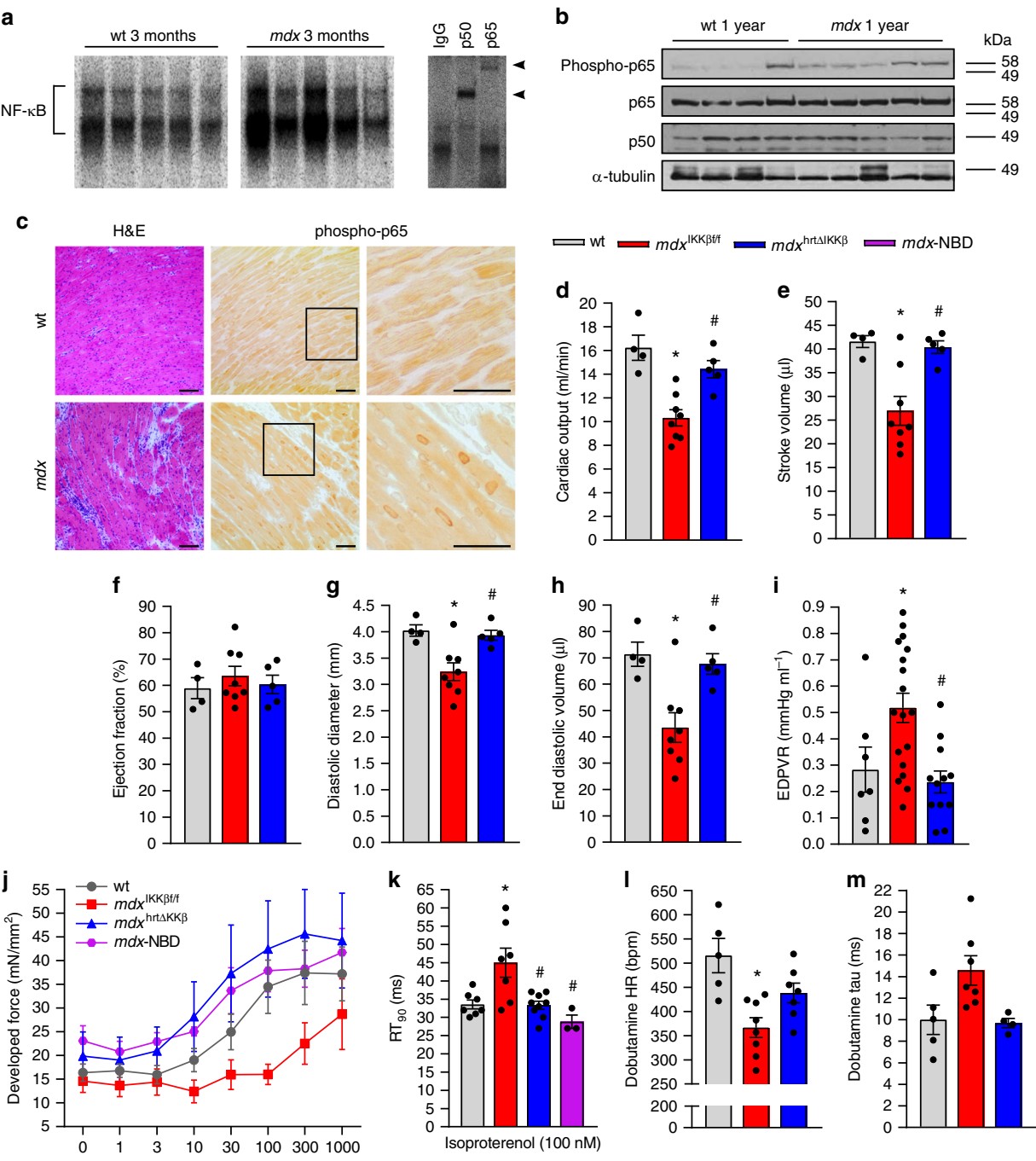

**Fig. 1** NF-κB causes heart dysfunction in *mdx* mice. **a** EMSA performed on wild-type (wt) and *mdx* hearts (left panel). Supershift EMSA performed on *mdx* hearts using specific antibodies for p65 and p50, and IgG as a control (Right panel). Arrowheads indicate shifted bands. **b** Western blots performed on whole heart lysates and probed for phosphorylated p65-ser536 (phospho-p65), p65, p50, and α-tubulin (used as a loading control). **c** Representative images of H&E and phosho-p65 staining prepared from 1-year old heart sections. Boxed regions appear as magnified images in neighboring panels. Scale bar = 50 μm. **d** cardiac output, **e** stroke volume, **f** ejection fraction ($p = 0.686$), **g** end diastolic diameter, and **h** end diastolic volume assessed by echocardiogram on 13–14-month-old mice ($n = 4$ wt; 8 $mdx^{IKKβf/f}$; 5 $mdx^{HRTΔIKKβ}$). **i** End-diastolic pressure volume relationship (EDPVR) assessed by ventricular pressure-volume relationship analysis on 13–14-month old mice ($n = 7$ wt; 18 $mdx^{IKKβf/f}$; 12 $mdx^{HRTΔIKKβ}$). **j** Developed force measured from isolated multicellular cardiac muscles of 7-month old mice in response to β-adrenergic stimulation with isoproterenol ($n = 7$ wt; 7 $mdx^{IKKβf/f}$; 9 $mdx^{HRTΔIKKβ}$; 3 NBD treated (*mdx*-NBD); $p = 0.278$). **k** Relaxation time ($RT_{90}$) after isoproterenol stimulation in multicellular cardiac muscles ($n =$ same as J). **l, m** Ventricular pressure-volume relationship measurements after dobutamine administration on 13–14-month-old mice. **l** Maximal heart rate (HR) ($n = 5$ wt; 8 $mdx^{IKKβf/f}$; 7 $mdx^{HRTΔIKKβ}$) and (**m**) Tau (isovolumetric relaxation) ($n = 5$ wt; 7 $mdx^{IKKβf/f}$; 4 $mdx^{HRTΔIKKβ}$; $p = 0.027$ but multiple comparisons test did not detect differences between groups). Data expressed as means ± SEM with bars and plungers and individual data points with dots. *$p < 0.05$ relative to wt and #$p < 0.05$ relative to $mdx^{IKKβf/f}$ by **d–i**, **k–m** 1-way ANOVA followed by Tukey multiple comparison test where appropriate, **j** 2-way repeated measures ANOVA. Main effects for genotype/treatment

| Table 1 Data are expressed as means ± SEM. 1-way ANOVA followed by Tukey multiple comparison test | | | | |
|---|---|---|---|---|
| Parameters | wt | *mdx*$^{IKK\beta f/f}$ | *mdx*$^{HRT\Delta IKK\beta}$ | P value |
| General | $n = 6$ | $n = 8$ | $n = 8$ | |
| Body weight (g) | 36.1 ± 0.7 | 33.5 ± 1.1 | 31.1 ± 1.5[a] | 0.0376 |
| Heart weight/tibia length (mg mm$^{-1}$) | 7.5 ± 0.1 | 6.6 ± 0.4 | 6.1 ± 0.2 [a] | 0.0073 |
| Echocardiogram | $n = 4$ | $n = 8$ | $n = 5$ | |
| HR (BPM) | 389.8 ± 17.7 | 396.1 ± 21.1 | 357.0 ± 13.1 | 0.3666 |
| Systolic diameter (mm) | 2.8 ± 0.2 | 2.2 ± 0.2 | 2.7 ± 0.2 | 0.0403 |
| End systolic volume (ml) | 29.8 ± 4.6 | 16.5 ± 3.1 | 27.3 ± 3.8 | 0.0432 |
| Left ventricular mass (mg) | 136.2 ± 6.4 | 139.2 ± 4.7 | 142.8 ± 7.5 | 0.7911 |
| FS (%) | 31.0 ± 2.7 | 34.2 ± 2.8 | 32.0 ± 2.4 | 0.7249 |

[a]Indicates different from wt where multiple comparison test detected differences. HR, (heart rate); FS, (fractional shortening)

To more closely investigate how NF-κB signaling affects cardiac contractility, we performed left ventricular pressure-volume relationship analysis. End-diastolic pressure volume relationship was increased in *mdx*$^{IKK\beta f/f}$ compared to wt mice and restored to wt values in *mdx*$^{HRT\Delta IKK\beta}$ mice (Fig. 1i), whereas end systolic pressure volume relationship was similar between all groups (Supplementary Table 1). Additional functional parameters that were not significantly altered are also reported in Supplementary Table 1. These data further support our echocardiogram results, suggesting that NF-κB signaling in *mdx* cardiomyocytes promotes diastolic dysfunction.

We next examined ex vivo cardiac function. We systemically treated 6-month-old *mdx* mice with NBD peptide for 1 month and performed function on multicellular cardiac preparations. We chose this time because ex vivo histological and functional cardiac deficits begin to develop at this stage. Age matched mice from our genetic model were used for comparisons. Although force values derived from *mdx*$^{IKK\beta f/f}$ mice were consistently lower across all frequencies compared to wt, these differences did not reach statistical significance (Supplementary Fig. 1D). In comparison, force production was higher in both *mdx*$^{HRT\Delta IKK\beta}$ and NBD-treated *mdx* muscles than in *mdx*$^{IKK\beta f/f}$ and wt muscles. We then assessed the ability of these multicellular muscles to relax, which affects cardiac diastole. Time from peak force to 90% relaxation (RT$_{90}$) trended toward being slower in *mdx*$^{IKK\beta f/f}$ muscles, whereas *mdx*$^{HRT\Delta IKK\beta}$ and *mdx*-NBD muscles showed relaxation times more similar to wt rates (Supplementary Fig. 1E). One indicator of a failing heart is the inability to respond to β-adrenergic stimulation[33]. We tested if this was dependent on NF-κB signaling. When cardiac muscles were challenged with increasing doses of the β-adrenergic stimulant isoproterenol, *mdx*$^{IKK\beta f/f}$ cardiac preparations responded with weak force production characteristic of their dystrophic phenotype (Fig. 1j)[26]. These mice were unresponsive to both low and moderate isoproterenol doses, only responding with slightly increased force production at high doses. In contrast, *mdx*$^{HRT\Delta IKK\beta}$ and NBD-treated *mdx* cardiac preparations responded to low isoproterenol doses following a parabolic curve similar to wt muscles. When we assessed relaxation, in response to isoproterenol, time from peak force to 90% relaxation (RT$_{90}$) was slower in *mdx*$^{IKK\beta f/f}$ muscles, whereas *mdx*$^{HRT\Delta IKK\beta}$ and NBD-treated *mdx* cardiac preparations showed an improved ability to accelerate their relaxation time back to wt levels (Fig. 1k). Together, these results indicate that pharmacological inhibition of NF-κB for just 1 month can reverse existing, stress induced cardiac muscle dysfunction, and that inhibition of NF-κB systemically and specifically in cardiomyocytes reverses the pathological inability of cardiac muscle preparations to respond to β-adrenergic stimulation.

To further explore the inability of *mdx* mice to respond to β-adrenergic stimulation, and the dependency of this response on cardiomyocyte NF-κB, we challenged mice with the β-adrenergic agonist dobutamine while performing pressure-volume loop analysis in vivo. Wt mice responded with a robust increase in heart rate, while *mdx*$^{IKK\beta f/f}$ mice were unable to increase their heart rates (Fig. 1l). *mdx*$^{HRT\Delta IKK\beta}$ mice responded with an increased heart rate, but did not normalize to wt. Isovolumetric relaxation time (tau) indicated that accelerated relaxation occurred in wt and *mdx*$^{HRT\Delta IKK\beta}$, but not in *mdx*$^{IKK\beta f/f}$ mice, although this did not reach significance (Fig. 1m). These data demonstrate that *mdx* cardiac muscle have an impaired ability to relax and are therefore unable to respond to β-adrenergic stress with heart rate acceleration. Cumulatively, data show that cardiomyocyte NF-κB impairs cardiac response to β-adrenergic stress, thus providing the first evidence that cardiomyocyte-derived classical NF-κB signaling plays an instrumental role in promoting dystrophic cardiac dysfunction. Although we cannot exclude the possibility that alternative NF-κB is compensating for the loss of IKKβ and contributing to the improved phenotype, processing of p100 to p52, as a measure of alternative pathway activation, was undetectable in the hearts of any of our mice (Supplementary Fig. 1F).

**Cardiomyocyte NF-κB inhibition improves calcium handling in *mdx* hearts.** Because cardiac stiffness from fibrosis leads to diastolic dysfunction, and expression of fibrotic genes is increased at an early age in *mdx* hearts (Supplementary Fig. 2A), we asked if NF-κB promoted cardiac fibrosis. Histologically, fibrosis was comparable in hearts from 1-year-old *mdx*$^{IKK\beta f/f}$ and *mdx*$^{HRT\Delta IKK\beta}$ mice (Supplementary Fig. 2B, C). This finding was consistent with gene expression analysis of the same panel of fibrotic genes that were elevated in *mdx* mice (Supplementary Fig. 2D). Membrane disruption also appeared similar between groups (Supplementary Fig. 2B). These data led to an intriguing implication that while cardiomyocyte NF-κB is not required for development of cardiac fibrosis or myocyte injury in *mdx* mice, it still contributes to cardiac dysfunction.

To gain insight into how NF-κB promotes cardiomyocyte dysfunction, we performed microarray analysis comparing *mdx*$^{HRT\Delta IKK\beta}$ to *mdx*$^{IKK\beta f/f}$ hearts. Gene Ontology (GO) and Kyoto Encyclopedia of Genes and Genomes (KEGG) pathway annotations identified genes related to calcium as enriched in the absence of NF-κB (Fig. 2a). Because of this proposed link between NF-κB and calcium, we examined calcium transient amplitudes at the single cell level to determine if gene expression changes coincided with functional outcomes. Results showed that calcium transients were reduced in *mdx*$^{IKK\beta f/f}$ compared to wt mice (Fig. 2b). In sharp contrast, calcium transient deficits were rescued in *mdx*$^{HRT\Delta IKK\beta}$ cardiomyocytes, indicating that NF-κB is intrinsically regulating calcium handling in *mdx* cardiomyocytes.

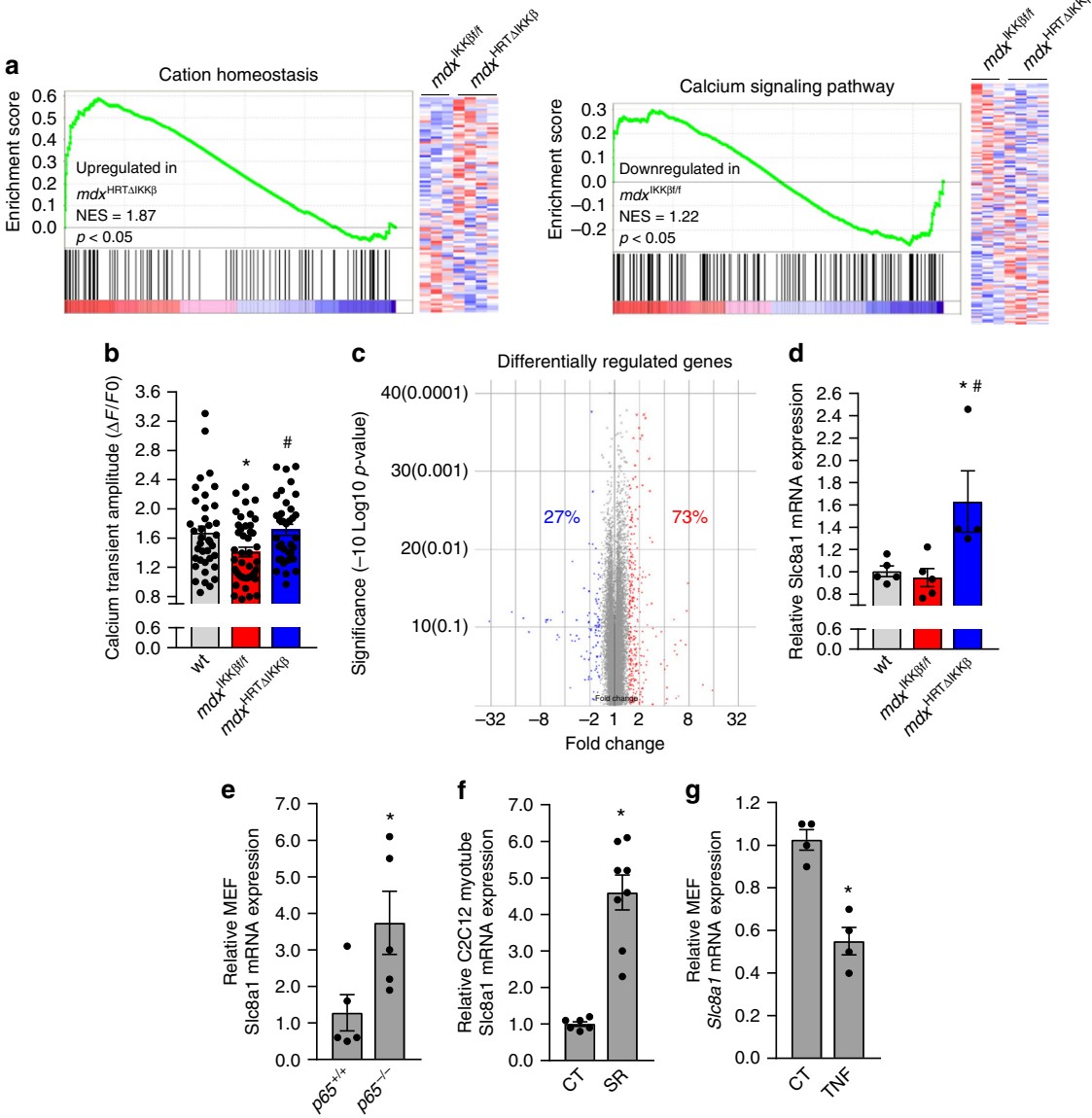

**Fig. 2** Cardiomyocyte NF-κB ablation normalizes calcium handling and increases gene expression. **a** Statistically significant gene categories from microarray analysis identified using Gene Set Enrichment Analysis. Heatmaps represent genes identified in annotations. **b** Calcium transient amplitude measured from cardiomyocytes isolated from 7–8-month old mice ($n = 38$ wt; 43 $mdx^{IKK\beta f/f}$; 35 $mdx^{HRT\Delta IKK\beta}$ cardiomyocytes). **c** Depiction of individual microarray genes that were up- and down-regulated in $mdx^{HRT\Delta IKK\beta}$ relative to $mdx^{IKK\beta f/f}$ hearts. Genes shown in red are ≥ 1.5-fold upregulated and those in blue are ≥ 1.5-fold downregulated. **d–g** qPCR analysis of *Slc8a1* expression. RNA isolated from **d** 6–7-month-old hearts ($n = 5$ wt; 5 $mdx^{IKK\beta f/f}$; 4 $mdx^{HRT\Delta IKK\beta}$), **e** mouse embryonic fibroblasts (MEFs) that were wt ($p65^{+/+}$) or null ($p65^{-/-}$) for $p65$ ($n = 5$). **f** C2C12 myotubes expressing empty vector as control (CT) or IκBα super repressor (SR) ($n = 6$ CT; 8 SR), and **g** MEFs untreated (CT) or treated with TNF ($n = 4$). Data expressed as means ± SEM with bars and plungers and individual data points with dots. *$p < 0.05$ relative to respective control and #$p < 0.05$ relative to $mdx^{IKK\beta f/f}$ by **b, d** 1-way ANOVA followed by Tukey multiple comparison test and **e–g** 2-tailed Student's *t* test

We then took a broader look at the overall gene expression pattern of dystrophic hearts lacking NF-κB to gain a better understanding for how this factor may be regulating calcium. NF-κB dimers containing the p65 subunit are usually thought of as transcriptional activators. Thus, when bound to consensus DNA sequences, target gene expression is typically stimulated[12]. Accordingly, we anticipated an overall reduction in gene expression upon deleting NF-κB in cardiomyocytes. Surprisingly, only 27% of differentially regulated genes (using 1.5-fold cut-off) were downregulated when NF-κB was deleted, whereas 73% were upregulated (Fig. 2c). Consistently, the majority of calcium genes identified in the GO and KEGG annotations were also upregulated in $mdx^{HRT\Delta IKK\beta}$ hearts (Fig. 2a). This is in sharp

contrast to microarray results we recently reported from Ras transformed $p65^{-/-}$ mouse embryonic fibroblasts (MEFs), which as expected, showed a predominance of down-regulated genes (89%) compared to $p65^{+/+}$ cells[34], or previous results in TNF treated HeLa cells, which demonstrated a similar predominance of NF-κB dependent upregulated genes[35]. Thus, our findings reveal a previously unappreciated functional role for NF-κB as a global repressor in *mdx* hearts.

Next, we sought to understand how NF-κB functions to repress calcium genes. We observed that differential expression was subtle, suggesting that small changes from multiple genes, rather than a large change from a single gene, might be responsible for improvements in calcium handling in the absence of NF-κB. We

searched for candidate genes that were both upregulated in $mdx^{HRT\Delta IKK\beta}$ hearts and associated with calcium. One gene that fit these criteria was the solute carrier family 8 (sodium/calcium exchanger), member 1, (*Slc8a1* gene that codes for the NCX1 protein). NCX1 is a transmembrane channel protein that plays a role in maintaining calcium homeostasis in multiple cell types including muscle[36]. Quantitative RT-PCR confirmed *Slc8a1* upregulation in $mdx^{HRT\Delta IKK\beta}$ compared to $mdx^{IKK\beta f/f}$ hearts (Fig. 2d), suggesting this gene is negatively regulated by NF-κB. This regulation was not restricted to cardiac tissue, since similar upregulation occurred in $p65^{-/-}$ compared to $p65^{+/+}$ MEFs and C2C12 skeletal myoblasts stably expressing the IκBα mutant super repressor (SR) inhibitor of NF-κB (Fig. 2e,f). Conversely, whereas TNF typically activates NF-κB to induce gene expression, TNF treatment of both MEF and C2C12 cells significantly reduced *Slc8a1* expression (Fig. 2g and Supplementary Fig. 2E). This response was specific to p65, since *Slc8a1* expression was comparable between $p50^{+/+}$ and $p50^{-/-}$ MEFs (Supplementary Fig. 2F). Collectively, these results support that NF-κB represses *Slc8a1*.

**NF-κB repression of *Slc8a1* occurs through depletion of H3K27ac.** We next investigated the mechanism of *Slc8a1* repression by NF-κB. Our previous studies in skeletal muscle demonstrated that NF-κB repressed myofibrillar genes through activation of the Polycomb chromatin remodeling complex associated protein, YY1[37,38]. However, no significant difference in *Yy1* expression was detected between $mdx^{HRT\Delta IKK\beta}$ and $mdx^{IKK\beta f/f}$ hearts (Supplementary Fig. 3A), suggesting that NF-κB does not utilize YY1 to repress gene expression in cardiomyocytes. To explore the possibility of an alternative chromatin remodeling mechanism, we used the UCSC Genome Browser to examine the chromatin landscape of *Slc8a1*. Results identified a putative regulatory region within the first intron containing a CpG island (Fig. 3a). In addition to regulating genes through DNA methylation, CpG islands influence chromatin conformation. Accordingly, we found the CpG island embedded in a 3 kb region, which ENCODE ChIP sequencing (ChIP-seq) datasets also showed were enriched for numerous histone modifications. Based on this information, we examined whether epigenetic and/or chromatin-mediated mechanisms were regulating *Slc8a1* expression by treating MEFs with the pan histone deacetylase inhibitor (HDACi) Trichostatin A (TSA) alone, or in combination with the DNA methylation inhibitor 5-Aza-2′-deoxycytidine (5-Aza). We first confirmed these drugs were effectively preventing histone deacetylation and demethylating DNA, respectively, using the gene, *Wif1*, which is epigenetically silenced in MEFs. *Wif1* was highly induced by the dual treatment of TSA and 5-Aza (Supplementary Fig. 3B). With respect to *Slc8a1*, TSA increased expression, whereas 5-Aza alone had no effect (Fig. 3b). TSA in combination with 5-Aza did not result in an additional increase over TSA alone. Consistently, TSA increased *Slc8a1* expression in HL-1 cardiomyocytes and C2C12 myoblasts (Fig. 3c, d). Together, these results suggest that NF-κB regulates *Slc8a1* through a histone modification-dependent mechanism.

We next performed ChIP assays to look for differences in several histone marks. We examined the acetylation (ac) and trimethylation (me3) of lysine 27 on histone H3 (H3K27). H3K27ac and H3K27me3 antagonize one another, indicative of transcriptional activation and repression, respectively. Additionally, we looked for me3 of histone H3 on lysine 9 (H3K9me3), a mark conducive for transcriptional repression. ChIP results revealed that *Slc8a1* was enriched for H3K27ac in $p65^{-/-}$ compared to $p65^{+/+}$ MEFs (Fig. 3e), whereas a loss of enrichment was detected in both H3K27me3 and H3K9me3 (Fig. 3f,g).

H3K27ac enrichment was also detected in C2C12 SR compared to control myoblasts (Fig. 3h). Moreover, ChIP assays revealed that treatment of MEFs with TSA enhanced H3K27ac on *Slc8a1* (Fig. 3i). To assess the significance of our results, we performed in vivo ChIPs. Results showed H3K27ac was depleted in $mdx^{IKK\beta f/f}$ hearts, but was normalized in $mdx^{HRT\Delta IKK\beta}$ hearts (Fig. 3j). Together these data support that loss of NF-κB signaling in dystrophic cardiomyocytes leads to a more permissive chromatin conformation on *Slc8a1*.

**NF-κB inhibition promotes global H3K27ac enrichment.** To determine if H3K27ac enrichment on *Slc8a1* in the absence of NF-κB was a global event, we performed in vivo ChIP-seq for H3K27ac. Strikingly, ~10,000 more H3K27ac regions were detected in $mdx^{HRT\Delta IKK\beta}$ compared to $mdx^{IKK\beta f/f}$ hearts (22,498 vs 12,148), highlighting global transcriptional activation in the absence of NF-κB signaling, and corroborating our microarray data. Although there were ~10,000 more individual H3K27ac regions detected in $mdx^{HRT\Delta IKK\beta}$ hearts, the genomic landscape of both our samples showed that the largest proportion of bound regions (~77%) was within intragenic and distal intergenic regions, which are the expected regions for an enhancer mark such as H3K27ac (Fig. 4a). Further interrogation revealed less than 20% of H3K27ac binding was within 10 kb of transcriptional start sites (Fig. 4b), substantiating that enrichment was confined mostly to genomic enhancers. To characterize our H3K27ac ChIP-seq results in the context of publicly available data sets from the ENCODE Consortium, we examined the genome-wide coverage of H3K27ac, H3K4me1, and H3K36me3 from normal hearts in relationship to our *mdx* hearts. Whereas ChIP-seq fragment densities peaked around our data with H3K27ac and another enhancer mark, H3K4me1, such a spike was absent in fragment densities around peaks with H3K36me3 ChIP-seq data. Instead, consistently elevated fragment densities occurred with H3K36me3, an expected pattern since it marks exons throughout actively transcribed genes (Fig. 4c and Supplementary Fig. 4A). These data indicate the overall binding pattern we observe is consistent with those globally expected in normal hearts.

Because we found that NF-κB suppresses genome-wide histone acetylation, we suspected that NF-κB binding would be located near enriched H3K27ac regions. To test this, we analyzed the density of H3K27ac binding in our samples across peaks from a publicly available p65 ChIP-seq data set. Results showed that increased H3K27ac was detected surrounding p65 binding peaks in the $mdx^{HRT\Delta IKK\beta}$ hearts, even when samples were adjusted to equalize the number of reads per experiment (Fig. 4d; total reads and Supplemental 4B; equalized reads). Importantly, the slope of the peak was shifted in $mdx^{HRT\Delta IKK\beta}$ hearts, reflecting a change in binding around these regions. These subtle differences in the genomic landscape were expected because the observed phenotype was a change in the degree of H3K27ac, as opposed to a complete abolishment of this activation mark. We confirmed by ChIP that p65 binding was also detected on *Slc8a1* (Supplementary Fig. 4C). When differentially enriched H3K27ac regions from $mdx^{HRT\Delta IKK\beta}$ relative to $mdx^{IKK\beta f/f}$ hearts were queried for significant GO pathways, networks of calcium-related terms were prominent (Fig. 4e). These findings reinforced that in the absence of NF-κB activation, global H3K27ac is enhanced on calcium genes of *mdx* hearts.

To validate ChIP-seq results, we selected a subset of calcium genes that were also upregulated in our microarray. These genes were *Slc8a1*, *Rcan1* (regulator of calcineurin 1), *Cacna1h* (calcium voltage-gated channel subunit alpha1 H), and *Camk4* (calcium/calmodulin dependent protein kinase IV). We confirmed that expression of these genes was upregulated in $mdx^{HRT\Delta IKK\beta}$

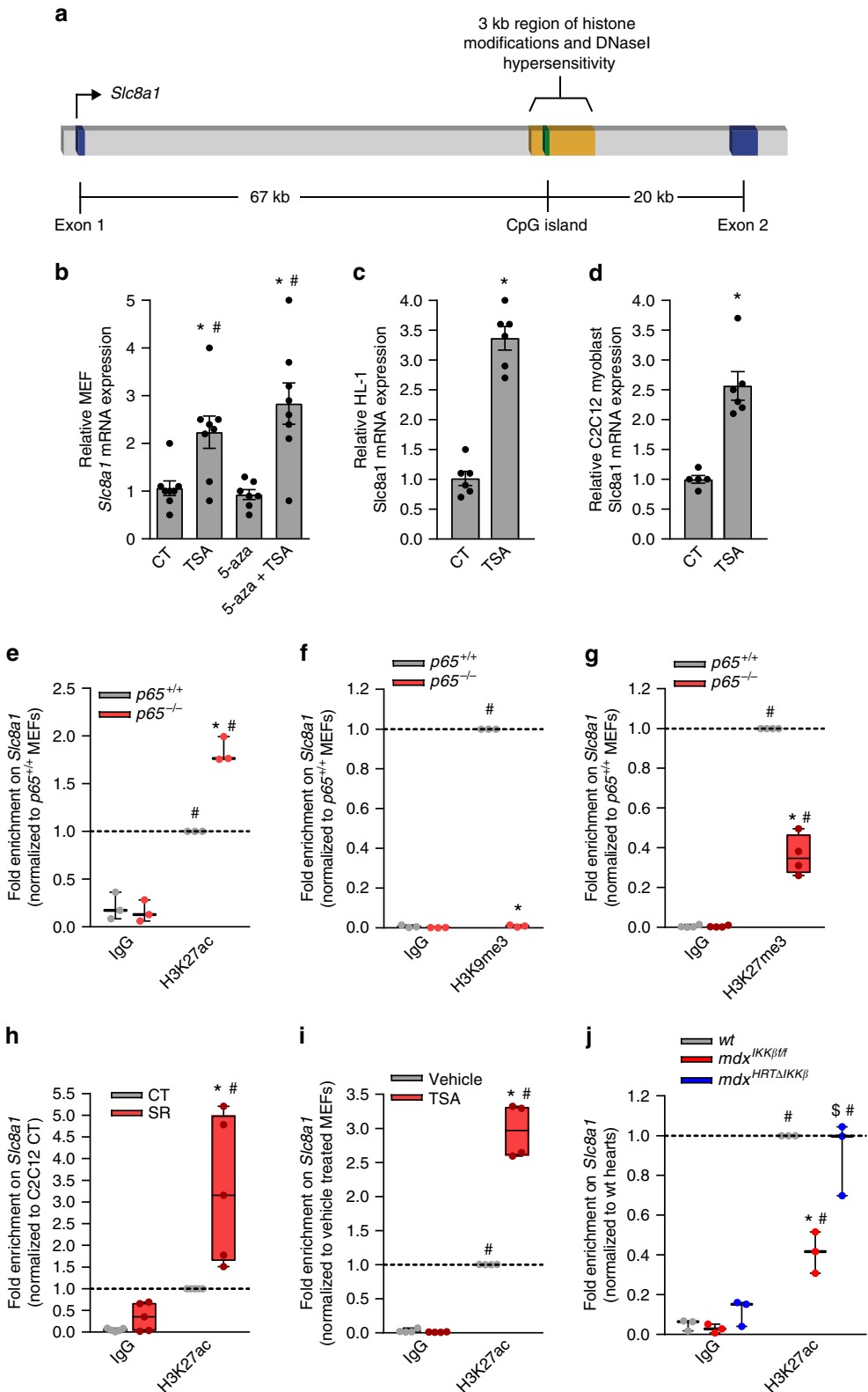

relative to $mdx^{IKK\beta f/f}$ hearts (Fig. 4f). Next, ChIP analysis on all four genes showed that H3K27ac was enriched in $mdx^{HRT\Delta IKK\beta}$ relative to $mdx^{IKK\beta f/f}$ hearts (Fig. 4g). To illustrate the dependency of H3K27ac on NF-κB activation, we treated MEFs with TNF. Treatment resulted in loss of H3K27ac enrichment (Fig. 4h). Interestingly, analysis of publicly available ChIP-seq data revealed that p65 peaks were present on all four genes near regions where we detected H3K27ac (Supplementary Fig. 4D).

Together, these results indicate that NF-κB orchestrates global repression of H3K27ac in $mdx$ hearts.

**NF-κB inhibition confers permissive chromatin through repressor redistribution.** To further explore the mechanism by which NF-κB inhibition regulates expression of calcium genes, we performed a motif analysis using our H3K27ac ChIP-seq from

**Fig. 3** NF-κB ablation increases *Slc8a1* through H3K27ac depletion. **a** Schematic depicting a regulatory region of interest within intron 1 of *Slc8a1*. Blue boxes = exons, orange box = the region of interest, green box = CpG island, arrow = TSS. **b–d** *Slc8a1* expression analyzed by qPCR on total RNA isolated from **b** MEFs that were vehicle treated (CT) or treated with trichostatin A (TSA), 5-Aza-2'-Deoxycytidine (5-aza), or a combination of TSA and 5-aza (n = 8 except 5-aza n = 7), and **c** HL-1 cardiomyocytes (n = 6) or **d** C2C12 myoblasts treated with vehicle (CT) or TSA and expressed relative to CT expression (n = 5 CT; 6 TSA). **e–j** ChIPs performed with **e, h–j** an H3K27ac antibody, (**f**) an H3K9me3 antibody (n = 3), or **g** an H3K27me3 antibody (n = 4) and qPCR analysis was used to detect enrichment in the *Slc8a1* regulatory region. DNA extracted from **e–g, i** MEFs (n = 3 **e**; 4 **i**), **h** C2C12 myoblasts (n = 5), or **j** mouse hearts (n = 3). **b–d** Data expressed as means ± SEM with bars and plungers and individual data points with dots. **e–j** Data expressed using box and whiskers plots. The central line in the boxes is the median value; the lower and upper boundaries of the boxes represent the lower and upper quartiles, respectively; the lower and upper whiskers represent the minimum and maximum values, respectively. Individual data points are plotted with dots.\*p < 0.05 CT, wt, or vehicle MEF, #p < 0.05 5-aza or IgG, and \$ p < 0.05 *mdx*^IKKbf/f by **b** 1-way ANOVA followed by Tukey multiple comparison test, by **c, d** 2-tailed Student's *t* test, or by **e–j** 2-way ANOVA followed by Tukey Post-hoc analysis

*mdx*^HRTΔIKKβ hearts. Due to the vast coverage of acetylated regions, it was difficult to determine other factors that may be interacting locally with NF-κB. We circumvented this issue by searching for motifs from a data set that included only genes containing both p65 peaks and H3K27ac regions. As expected, the motif containing binding sites for the NF-κB family of proteins (Rel Homology Domain; RHD) was the highest ranked motif observed (Fig. 5a). We grouped motifs into families where possible (representative core sequences shown in Fig. 5a) and searched for a candidate repressor protein. The highest ranked repressor motif was CCCTC-binding factor (CTCF). The CTCF protein that binds these motifs is capable of repressing gene expression[39]. To investigate if CTCF was associated with genes repressed in *mdx* hearts, we ran a filtered list of H3K27ac regions associated with genes that were de-repressed in *mdx*^HRTΔIKKβ hearts. 81% of the genes containing a CTCF motif were upregulated in *mdx*^HRTΔIKKβ hearts (Fig. 5b), indicating that CTCF may be involved in repressing gene expression upon NF-κB is activation.

Next, we analyzed publically available CTCF ChIP-seq data and compared it to the genome-wide coverage of p65 peaks to investigate the global relationship between p65 and CTCF. Impressively, p65 binding was closely associated with coverage of CTCF, demonstrating that p65 and CTCF are mutually bound to many gene loci (Fig. 5c). When CTCF fragment densities around NF-κB peak centers were evaluated based on the presence of NF-κB motifs, we found approximately 2-fold higher sequenced fragments from the CTCF data around peaks lacking canonical NF-κB binding sites (Fig. 5d). This more prominent association between CTCF and non-canonical p65 binding was not due to a greater number of total p65 peaks, since non-canonical peaks were still found near CTCF peaks after a random normalization for the number of peaks between the two groups was performed (Supplementary Fig. 5A).

ChIPs confirmed enriched occupancy of CTCF at sites close to the enriched H3K27ac region in *mdx* hearts on *Slc8a1, Rcan1, Cacna1h, and Camk4* (Fig. 5e). Importantly, CTCF occupancy was depleted in *mdx*^HRTΔIKKβ relative to *mdx*^IKKβf/f hearts, suggesting that CTCF binding was responsive to NF-κB inhibition. We then tested CTCF binding in response to NF-κB activation. Corroborating our results in hearts, TNF treatment enriched CTCF binding on all four genes (Fig. 5f). However, no reduction in CTCF binding was observed in *p65*^−/− compared to *p65*^+/+ MEFs (Supplementary Fig. 5B), indicating that CTCF binding does not require p65. These results suggest that in response to an inflammatory condition, CTCF regulates chromatin interactions on calcium genes, but this regulation is not directly mediated by NF-κB.

We then asked if repressor complexes known to reduce H3K27ac were affected by the loss of p65. In MEFs, SIN3 transcription regulator family member A (SIN3A) and histone deacetylase 1 (HDAC1) showed enrichment in the same regions we detected differences in H3K27ac on *Slc8a1, Rcan1, Cacna1h, and Camk4* (Supplementary Fig. 5C, D). Conversely, loss of enrichment occurred in the absence of p65. In *mdx*^IKKβf/f hearts, enrichment of SIN3A and HDAC1 was observed on calcium genes (Fig. 5g,h). Except for SIN3A on *Cacna1h*, this enrichment was diminished in *mdx*^HRTΔIKKβ hearts. These data suggest that p65 cooperates with chromatin repressors to reduce H3K27ac on calcium genes and limit their expression. To investigate if NF-κB was capable of antagonizing H3K27ac, we treated HL-1 cardiomyocytes with TSA and TNF. Enhanced gene expression afforded by TSA was diminished by TNF treatment (Fig. 5i). Such data support that upon activation, NF-κB is capable of antagonizing H3K27ac, even in the presence of an HDACi. We then asked if the binding of repressor complex members near p65 peaks was a global occurrence. Because our ChIP-seq was performed with H3K27ac, we used a p65 ChIP-seq data set. The overall binding density of SIN3A and HDAC1 was greater surrounding p65 peaks on genes upregulated in *mdx*^HRTΔIKKβ hearts (Fig. 5j). Together, these data indicate that in the presence of p65, repressive complexes occupy calcium genes, resulting in a less permissive chromatin conformation.

## Discussion

DMD is a devastating disease that adversely affects both skeletal and cardiac muscles. However, little is known regarding the mechanisms of cardiomyopathy, particularly when it comes to the contribution of individual signaling pathways. Although p38-MAPK is aberrantly activated in *mdx* hearts[40], the impact of this activation and the identity of other signaling pathways that may contribute to DMD cardiomyopathy have not been explored. Perhaps better understood, is that disruption of calcium home-ostasis is considered causative for the pathology of dystrophic hearts[10]. More specifically, a leaky sarcolemma due to the primary loss of dystrophin, activation of calcium leak channels, and alterations in ryanodine receptor channel function, have each been linked to altered calcium concentrations and cellular signaling. However, what causes the alterations in calcium handling proteins and whether reversing these changes rescues dystrophic cardiomyopathy remain unclear.

In this study, we show that inhibiting NF-κB alleviates cardiac dysfunction in *mdx* mice by orchestrating global chromatin remodeling to improve calcium signaling. In this respect, we showed that in cardiomyocytes, p65 functions in a different manner than its typical role of directly binding DNA and activating gene expression. Rather, in dystrophic hearts, NF-κB binds to both its consensus and non-consensus sites, which associates with the enrichment of CTCF on calcium genes. In these same regions, HDAC1 and SIN3A are also enriched, resulting in loss of H3K27ac and a less permissive chromatin conformation. In the absence of p65, HDAC1 and SIN3A binding are reduced, resulting in enriched H3K27ac and a more permissive chromatin

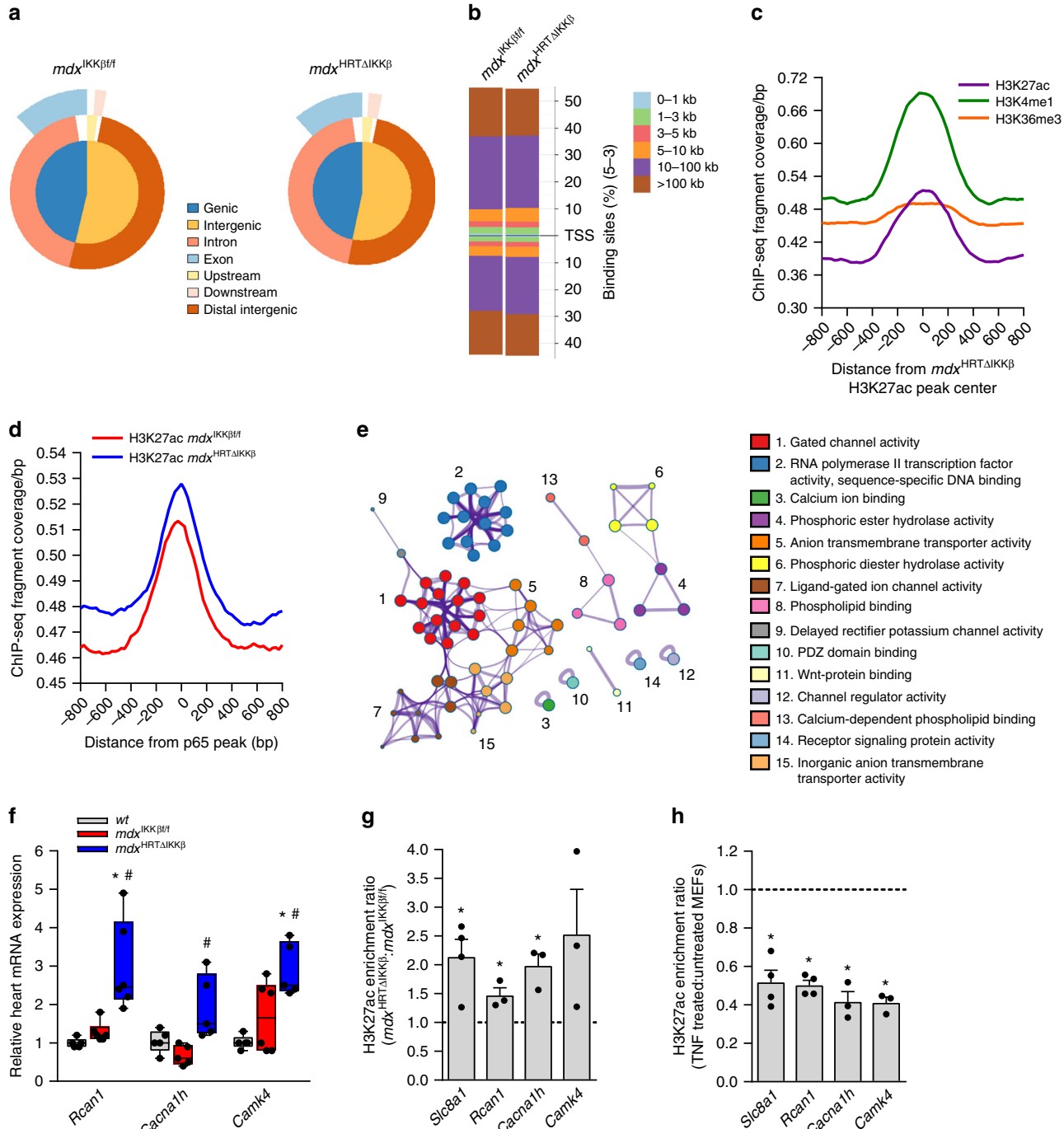

**Fig. 4** Cardiomyocyte NF-κB ablation causes global H3K27ac enrichment in *mdx* hearts. **a** Venn diagram pie chart depicting the H3K27ac ChIP-seq annotation within specified regions of the genome from mouse hearts. **b** Genome-wide distribution of H3K27ac binding loci relative to transcription start sites (TSS). **c–d** Genome-wide fragment density showing potential overlap of **c**. ChIP-seq histone marks across peaks from our ChIP-seq performed in *mdx*^HRTΔIKKβ hearts and **d** our ChIP-seqs across peaks from p65 ChIP-seq. **e** Network showing the top 15 Gene Ontology clusters identified from differentially enriched genes in the *mdx*^HRTΔIKKβ when compared to *mdx*^IKKβf/f H3K27 regions. The most significantly enriched pathway for each cluster is labeled as the representative term for that group. **f** Gene expression analyzed by qPCR on total RNA isolated from hearts (n = *Rcan1*: 5 wt; 6 *mdx*^IKKβf/f; 6 *mdx*^HRTΔIKKβ; *Cacna1h*: 5 for all genotypes; *Camk4* 5 wt; 6 *mdx*^IKKβf/f; 5 *mdx*^HRTΔIKKβ). **g–h** ChIP performed with an H3K27ac antibody and qPCR analysis was used to detect enrichment on denoted genes. DNA extracted from **g** mouse hearts (n = 3) or **h** control and TNF treated MEFs (n = 4 *Slc8a1* and *Rcan1* and 3 *Cacna1h* and *Camk4*). Genes were expressed as a ratio. Dotted line represents level of enrichment equal to **g** *mdx*^IKKβf/f hearts and **h** control MEFs. Bars represent (**g**) enrichment in hearts and **h** depletion in TNF treated MEFs. **f** Data expressed using box and whiskers plots. The central line in the boxes is the median value; the lower and upper boundaries of the boxes represent the lower and upper quartiles, respectively; the lower and upper whiskers represent the minimum and maximum values, respectively. Individual data points are plotted with dots. **g–h** Data expressed as means ± SEM with bars and plungers and individual data points with dots. **f** *p < 0.05 wt and # p < 0.05 *mdx*^IKKβf/f, by 1-way ANOVA followed by Tukey multiple comparison test. **g–h** *p < 0.05 *mdx*^IKKβf/f or untreated MEFs by 2-tailed Student's *t* test

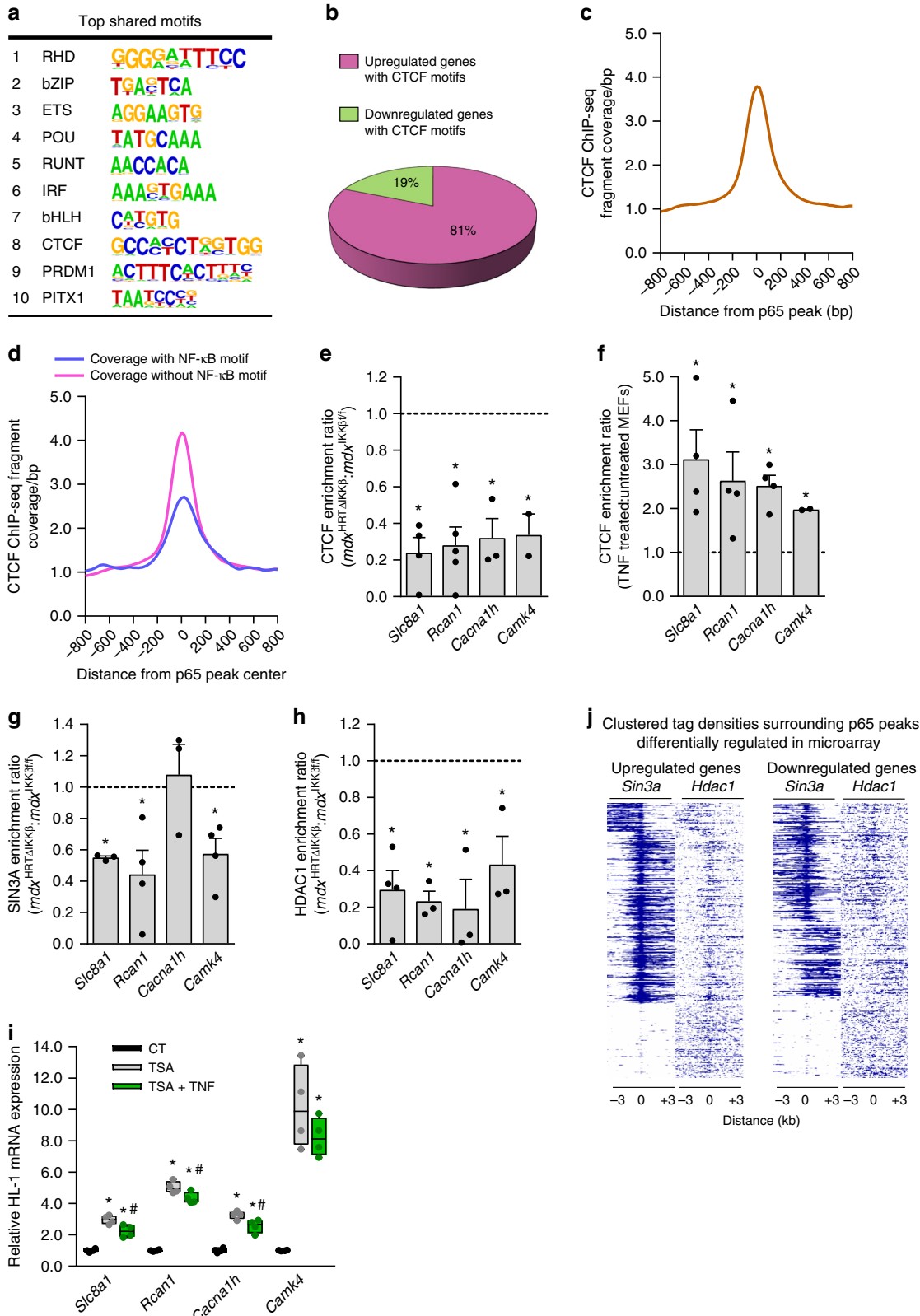

conformation. Thus, in DMD, NF-κB inhibition alters the enhancers of calcium homeostatic genes, which normalizes cardiac function.

NF-κB has previously been shown to interact directly with HDAC1 where it can deacetylate p65, functionally reducing its ability to activate gene expression[41]. Our data suggest that NF-κB also interacts with HDAC1 to assist in the deacetylation of

chromatin on calcium homeostatic genes. In addition to HDAC1, we found that p65 interacts with SIN3A (Supplementary Fig. 5E). This suggests that p65, SIN3A, and HDAC1 form a complex that is responsible for the deacetylation of H3K27, and ultimately for the transcriptional repression of calcium genes. These observations have exciting implications, as HDACi are being actively pursued as a therapeutic for treating skeletal muscle pathology in

**Fig. 5** CTCF, SIN3A, and HDAC1 mediate a less permissive chromatin conformation on calcium genes upon NF-κB activation. **a** Motif analysis performed on genes identified as having both p65 ChIP-seq peaks and H3K27ac $mdx^{HRTΔIKKβ}$ ChIP-seq regions. **b** Pie graph representing the percentage of genes with CTCF motifs up- and downregulated in the microarray (relative to $mdx$ hearts with intact NF-κB). **c** ChIP-seq data derived from genome-wide fragment density analysis showing potential overlap of CTCF peaks with p65 peaks. **d** The same analysis as in **c** except p65 peaks were split between two groups either containing or lacking an NF-κB consensus motif. **e–f** ChIP performed with a CTCF antibody and qPCR analysis was used to detect enrichment on denoted genes. DNA extracted from **e** $mdx^{IKKβf/f}$ and $mdx^{HRTΔIKKβ}$ hearts ($n = 4$ Slc8a1; 5 Rcan1; 3 cacna1h; 2 Camk4) and (**f**) control and TNF treated MEFs ($n = 4$ except $n = 2$ Camk4), **g–h** The same analyses were performed as in **e**, **f**. DNA extracted from $mdx^{IKKβf/f}$ and $mdx^{HRTΔIKKβ}$ hearts and ChIP performed with a (**g**) SIN3A antibody ($n = 3$ Slc8a1 and Cacna1h ($p = 0.7$); $n = 4$ Rcan1 and Camk4) (**h**) HDAC1 antibody ($n = 3$ except $n = 4$ Slc8a1). **e–h** Enrichment on different genes were plotted as a ratio. Dotted line represents level of enrichment equal to **e, g–h** $mdx^{IKKβf/f}$ hearts and (**f**) control MEFs. Bars represent **e, g–h** depletion in $mdx^{HRTΔIKKβ}$ hearts and **f** enrichment in TNF treated MEFs. **i** Gene expression analyzed by qPCR on total RNA isolated from HL-1 cardiomyocytes ($n = 4$). **j** Heatmaps showing ChIP-seq fragment densities of SIN3A and HDAC1 surrounding p65 peaks, showing potential overlap. Left panel includes genes upregulated and right panel includes genes downregulated in the microarray. **e–h** Data expressed as means ± SEM with bars and plungers and individual data points with dots. (**i**) Data expressed using box and whiskers plots. The central line in the boxes is the median value; the lower and upper boundaries of the boxes represent the lower and upper quartiles, respectively; the lower and upper whiskers represent the minimum and maximum values, respectively. Individual data points are plotted with dots. **e–h** *$p < 0.05$ $mdx^{IKKβf/f}$ or untreated by 2-tailed Student's $t$ test. **i** *$p < 0.05$ CT and #$p < 0.05$ TSA by 1-way ANOVA followed by Tukey multiple comparison test

DMD[42]. Although HDACi have not been tested for their ability to improve heart function per se, they have been shown to reduce cardiac arrhythmias in $mdx$ mice[43]. In addition, a recent study reported that administration of an HDACi to diabetic mice resulted in improved cardiac function[44], implying that if HDACi are approved for the treatment of skeletal muscle pathology in DMD boys, they may have the added benefit of improving heart function.

It is noteworthy that all four of the calcium homeostatic target genes we examined were transcriptionally active in wt hearts. This demonstrates that none of these genes are epigenetically silenced in cardiac muscle, as we anticipated, since their expression contributes to cardiac homeostasis. We made an interesting observation when we compared the expression levels of these calcium genes between wt and $mdx^{IKKβf/f}$ hearts. Although NF-κB is activated in dystrophic hearts, we find that NF-κB is not acting in its canonical role as a direct transcriptional activator, but rather as a modulator of chromatin conformation to deplete H3K27ac. We anticipated that depletion of H3K27ac would result in reduced expression of these calcium target genes, but this was not the case. We speculate that this absence of repression could be due to a compensatory response by a dystrophic heart to activate these genes in attempt to maintain calcium homeostasis and normal heart function. For example, transcriptional regulation of $Slc8a1$ was previously attributed to MAP kinases[45], which are also activated in $mdx$ hearts[40,46]. Thus, it is possible that while MAP kinase signaling is actively attempting to upregulate $Slc8a1$ transcription, activation of NF-κB is conferring a restricted chromatin conformation, thus limiting access to transcription factors downstream of MAP kinase activity that are needed to stimulate higher $Slc8a1$ expression. Therefore, with NF-κB inhibition, local chromatin regains a more permissive conformation, allowing access to transcription factors so that a compensatory response occurs and cardiac function is maintained at wt levels. It will be interesting in future studies to address whether the ability of NF-κB to regulate H3K27ac and restrict chromatin confirmation occurs under physiological conditions in normal cardiomyocytes, or as shown with C2C12 and MEFs, is relevant in other non-cardiac tissues.

Although NF-κB has not been studied in depth in the context of dystrophic cardiomyopathy, its association in various other models of heart disease has been documented. For example, when NEMO was conditionally deleted from non-diseased mouse hearts, this resulted in aged related dilated cardiomyopathy and fibrosis that could be elicited in younger mice using a pressure overload model[47]. These data are in stark contrast to our findings where we observed improved function with no difference in fibrosis. These differences could be attributed to deletion of IKKβ vs NEMO, or potentially due to the difference in phenotype when NF-κB signaling is interrupted in a disease state (dystrophy) rather than in a normal condition. In heart failure models, inhibiting NF-κB has beneficial effects including improved survival and function, and reduced apoptosis and hypertrophic remodeling[48–50]. Additionally, one report showed that cardiomyocyte $p65$ deletion preserved calcium handling after ischemia-reperfusion injury[51]. This improvement corresponded with increased phosphorylation of serine 16 on phospholamban, resulting in increased sarcoplasmic reticulum calcium uptake. We too observed increased phosphorylation on phospholamban, but perceive this post-translational modification as a downstream event rather than a direct result of $p65$ deletion since NF-κB subunits do not possess phosphatase activity. NF-κB inhibition also prevented the development of diabetic cardiomyopathy[52]. Mice were reported to maintain normal calcium handling with no aberrant activation of the renin-angiotensin pathway. These findings are supportive of our conclusion that NF-κB signaling is detrimental to the stressed heart, and that inhibiting this pathway provides beneficial outcomes.

Like NF-κB, the functions of CTCF are vast, and sometimes contradictory. They include maintaining higher order chromatin structure, gene insulation, maintaining borders between euchromatin and heterochromatin, X-inactivation, gene repression, and gene activation[39]. We show that when NF-κB is activated, CTCF occupies its binding sites, conferring a higher order chromatin conformation on calcium genes that favor a repressive structure. CTCF occupancy was not reduced in the absence of p65, indicating that although CTCF is responsive to NF-κB activation, its binding does not require the co-binding of p65. Higher order chromatin conformation changes due to CTCF binding has previously been shown to occur in response to inflammatory induced NF-κB activation, which is similar to what we have shown, but how this occurred was not elucidated[53]. One group reported that CTCF is a direct transcriptional target of NF-κB[54]. Although this could explain why in our study CTCF binding was reduced in the absence of NF-κB activation, in our hands CTCF mRNA expression was unchanged in $mdx^{HRTΔIKKβ}$ hearts or in $p65^{-/-}$ compared to $p65^{+/+}$ MEFs. It therefore remains to be determined how CTCF is regulated on calcium genes in response to NF-κB activation.

We performed this study because there is a need to find effective therapeutics whose mechanism of action is understood, and whose functions are capable of targeting both cardiac and skeletal muscle in DMD. Through our previous genetic studies, we identified mechanisms for how NF-κB signaling is detrimental

to the skeletal muscles of dystrophic mice and then showed that administration of the NBD peptide improves skeletal muscle pathology and function in both dystrophic mice and canines. We now add to these findings by showing a mechanism for how NF-κB signaling is detrimental to dystrophic cardiac function. To our knowledge, NF-κB represents the first signaling pathway capable of promoting pathology in both dystrophic cardiac and skeletal muscle. Understanding its mode of action in both muscle types now provides stronger rationale for targeting this pathway as a therapeutic for DMD.

## Methods

**Mice**. Animal use and experimentation for this study was approved by The Ohio State University (OSU) Institutional Animal Care and Use Committee and all experiments were performed in accordance with relevant ethical guidelines and regulations. *Mdx* mice were originally purchased from The Jackson Laboratory (C57BL/10 ScSn DMD*mdx*) and have been housed and bred in the animal facilities of OSU for 10 years. Myh6 cre and IKKβ flox mice were previously created[55,56]. Mice were kept on a 14:10 light-dark cycle with constant temperature and humidity and fed a standard diet. Male mice were used for these studies.

**Cell culture**. MEFs were previously isolated and immortalized in our lab from *p65+/+*, *p65−/−*, *p50+/+*, and *p50−/−* mice[57]. C2C12 mouse myoblast cell line and C2C12 cells stably expressing a mutant IκBα plasmid (C2C12 SR) were utilized[58]. Cells were originally obtained from ATCC and cultured in high-glucose DMEM containing 10% fetal bovine serum and antibiotics. C2C12 myoblasts were differentiated in high-glucose DMEM without sodium butyrate supplemented with 2% horse serum, 100 ng ml$^{-1}$ insulin, and antibiotics. HL-1 cardiomyocytes were obtained from SIGMA and cultured in Supplemented Claycomb Medium. Cells were treated with 10 ng ml$^{-1}$ TNFα for 4 h (ChIP and gene expression with TSA treatment) or 1 ng ml$^{-1}$ for 2 days (gene expression with no treatment), 50 nM TSA for 24 h (MEF and C2C12) or 48 h (HL-1), 1 μM 5-Aza for 1 passage. For combination experiments, 5-Aza was added for one passage then MEFs were treated with TSA for 24 h. For TSA + TNF, cells were treated with TSA for the indicated time (see above) and then treated with TNF for the last 4 h of experiment. Cells lines were not authenticated or tested for mycoplasma.

**NBD mouse administration**. NBD peptide was administered by intraperitoneal injections[18]. Injections were performed three times per week for 1 month starting at 6 months of age at a dose of 10 mg kg$^{-1}$.

**EMSA**. Electrophoretic mobility shift assay (EMSA) and supershifts were performed using a standard protocol[59]. Hearts were homogenized and nuclear extracts prepared. Extracts were incubated with a radioactive oligonucleotide containing a consensus NF-κB binding site and fractionated on a 5% non-denaturing polyacrylamide gel. For supershifts, extracts were incubated with the following antibodies: 4 μl p50 (114 Santa Cruz) or IgG or 0.5 μl p65 (Rockland) prior to loading on the gel.

**Western blotting**. Protein was extracted and immunoblotting procedures performed. Primary antibodies: phosphorylated p65 serine 536 (1:1,000, Cell Signaling 3031), p65 (1:10,000; Rockland, non-catalog item), p50 (Santa Cruz, sc-114). Uncropped scans are available in Supplementary Fig. 6.

**Tissue preparation and histology**. Hearts were either snap frozen for RNA and protein or mounted in OCT and frozen in 2-Methylbutane cooled to the temperature of liquid nitrogen. 10 μm sections were prepared on a cryostat. Histological staining was performed for H&E. Immunohistochemistry was performed using a phosphorylated p65 serine 536 (1:500; Cell Signaling, 3031) primary antibody, HRP goat anti-rabbit IgG (1:250, Vector, BA-1000) secondary antibody and a DAB substrate kit (Vector, SK-4100) for detection.

**Multicellular cardiac contractile function**. Cardiac muscles were isolated and baseline measures, frequency-dependent activation, and β-adrenergic stimulation were accessed ex vivo. Under deep anesthesia, hearts were rapidly removed, and flushed with a Krebs-Henseleit solution. The right ventricle was opened, and small papillary muscles were dissected under a stereo microscope. Muscles were mounted in an experimental chamber and superfused with Krebs-Henseleit solution, containing 1.5 mM Ca$^{2+}$, at 37 °C. Muscles were electrically stimulated to twitch contract, and force of contraction was recorded. For baseline measurements, after the muscle had equilibrated in the set-up, muscle length was increased until a further increase in length no longer resulted in an increase in active twitch developed peak force. This length was then considered optimal length. Frequency-dependent stimulation (between 4 and 14 Hz), and β-adrenergic stimulation (1 nM – 1000 nM Isoproterenol) responses were then determined. Additionally, as a

model-independent parameter of force decay kinetics, time from peak force to 90% relaxation (RT$_{90}$), was determined.

**Echocardiography**. Echocardiography was performed using a VEVO 2100 Visual Sonics (VisualSonics, Toronto) system. Mice were lightly anesthetized (1.5% iso-flurane) and ejection fraction, fractional shortening, and ventricular chamber dimensions were measured in M mode using the parasternal short axis view.

**Left ventricular pressure-volume relationship analysis**. Cardiac hemodynamic measurements were assessed via a closed chest approach using a 1.4 French Millar Pressure catheter (AD Instruments) advanced into the left ventricle through the right carotid artery. Mice were anaesthetized by ketamine (55 mg kg$^{-1}$) plus xylazine (15 mg kg$^{-1}$). After 5–10 min of stabilization, values at baseline and stimulation at varying frequencies (4–10 Hz) were recorded. Pressure volume loops at 7 Hz were also obtained at varying preloads via inferior vena cava occlusions to get EDPVR, ESPVR, and PRSW. To measure the beta-adrenergic response, 5 mg kg$^{-1}$ dobutamine was injected intraperitoneal. All the measurement and analysis were performed on LabChart7 (AdInstruments).

**Gene expression analysis**. Total RNA was isolated using a standard Trizol extraction protocol. 1 μg of RNA and M-MLV reverse transcriptase was used for cDNA synthesis. Quantitative Real-time PCR was performed using SYBR Green on either an Applied Biosystems StepOnePlus or BioRad CFX96 Touch system. All data were normalized to *gapdh* and the ΔΔct method was applied for analysis. Primer sequences are included in Supplementary Table 2.

**Microarray**. Total RNA was isolated using a standard Trizol extraction protocol and purified with a cleanup kit (Qiagen). A GeneChip Mouse Genome 430 2.0 Affymetrix array was used. Gene summary expression estimates were retrieved using Robust multi-array average (RMA) method from probe level data after back ground correction and quantile normalization with Partek software (http://www.r-project.org/). RMA and gene expression data were also independently obtained using Expression Console software from Affymetrix. Gene Set Enrichment Analysis[60,61] was used for further analysis.

**Cardiomyocyte isolation**. Cardiomyocytes were isolated by Langendorff perfusion. Hearts were cannulated and perfused with Ca$^{2+}$ free tyrode solution (Normal tyrode solution in mM: 140 NaCl, 4 KCl, 1 MgCl$_2$, 10 glucose, and 5 HEPES, pH 7.4 adjusted with NaOH or HCl) for 5 min. Subsequently, hearts were perfused with a tyrode solution containing Liberase Blendzyme II (0.077 mg ml$^{-1}$) (Roche Applied Science, Indianapolis, IN). After 4–6 min, hearts were removed from the Langendorff apparatus, the ventricles minced, and myocytes dissociated via trituration. Myocytes were then filtered and centrifuged before being resuspended in a tyrode solution containing 200 μM Ca$^{2+}$.

**Myocyte calcium transient measurements**. Myocytes were loaded with Flou-4 Am (10 μM, Molecular Probes, Eugene, OR) and incubated for 30 min. Myocytes were then washed and allotted an additional 30 min for de-esterification. A Cairn Research Limited (Faversham, UK) epifluorescence system was used for intracellular Ca$^{2+}$ measurements (Fluo-4 epifluorescence with excitation: 480 ± 20 nm and emission: 535 ± 25 nm). The change in fluorescent intensity is expressed as ΔF/F$_0$, where *F is the* fluorescence intensity and $F_0$ is the fluorescence intensity at rest.

**ChIP**. Cells were fixed with 1% formaldehyde for 8 min at 37 °C. Formaldehyde was quenched with 1/20 volume of glycine (2.5 M), cells washed with PBS, and scraped in nuclei isolation buffer (20 mM PIPES (KOH) pH 8.0, 85 mM KCl, 0.5% NP-40, 1x protease inhibitors). Nuclear pellets were lysed (10 mM EDTA, 50 mM Tris-HCl pH 8.0, 1% SDS, 1x protease inhibitors) and sonicated with a probe sonicator. Supernatant was precleared with tRNA/Protein G agarose slurry in buffer (1.2 mM EDTA, 16.7 mM Tris-Hcl pH8.0, 167 mM NaCl, 1.1% Triton-x 100, 0.01% SDS, 1x protease inhibitors) for 2 h then incubated overnight using the following antibodies from Millipore: 3 μg H3K27ac (17–683 and 07–360), HDAC1 (17–10199), and IgG (Rabbit, 12–370; Mouse, 12–371), Active Motif: 5 μg H3K9me3 (61013) and H3K27me3 (39155), Novus: 6 μg SIN3A (600–1263), Santa Cruz: 3 μg CTCF (5916). For tissue ChIP, sodium heparin was administered to mice prior to excising heart. Hearts were washed in PBS, minced, then simultaneously fixed and homogenized (1% formaldehyde in nuclei isolation buffer) for a total time of 15–20 min at 4 °C. Nuclei were washed with PBS and then lysed and sonicated as described in cells. Supernatants were precleared for 4 h and ChIP proceeded as described in cells with the addition of a centrifugation step following overnight antibody incubation. Quantitative Real-time PCR was performed using TaqMan (Roche) on a BioRad CFX96 Touch system. Primer sequences are included in Supplementary Table 3.

**ChIP-sequencing**. ChIP was performed with a 15 μg H3K27ac antibody (17–683, Millipore). Purified DNA (10 ng) was end-repaired, and A-nucleotide overhangs were added by incubation with the Taq Klenow fragment lacking exonuclease activity. After the attachment of anchor sequences, fragments were PCR-amplified

using Illumina-supplied primers. The purified DNA library products were evaluated using Bioanalyzer (Agilent) and SYBR qPCR and diluted to 10 nM for sequencing on HiSeq 2000 sequencer (pair-end with 50bp). A data analysis pipeline SCS v2.5 (Illumina) was employed to perform the initial bioinformatics analysis including base calling and converting the results into raw reads in FASTQ format.

**ChIP-sequencing analysis.** Bowtie2 was used to map uniquely aligned sequence reads to the mouse reference genome (mm9)[62]. HOMER (Hypergeometric Optimization of Motif EnRichment) software suite was used to create tag directories, for peak calling and annotation, for motif analysis, to create histograms, determine differentially enriched peaks, and generate a data matrix for heatmaps[63]. ChIP-seeker software was used to visualize genomic regions of H3K27ac binding and binding relative to TSS[64]. Metascape was used to identify pathway enrichment in differential peaks [http://metascape.org][65]. Cytoscape was used to visualize the clustered pathways[66]. Cluster 3.0[67] was used to cluster the data matrix files and Java TreeView[68] was used to visualize these data in heatmaps.

**Publicly available data.** The following publicly available ChIP-seq data sets were used. 8-week old hearts were from the ENCODE Consortium produced by the Ren lab[69]. H3K27ac (GSM1000093), H3K4me1 (GSM769025), H3K36me3 (GSM1000130), CTCF (GSM918756). p65 was from LPS treated bone marrow derived macrophages (GSM611116; GSM611117). Our previous microarray was in Ras transformed MEFs (GSE59545).

**Statistics.** All data were expressed as mean ± SEM. Results were analyzed by an unpaired 2-tailed Student's t test, 1-way ANOVA, or repeated measures 2-way ANOVA as appropriate. Data met the requirements for performing the appropriate test. A P value of less than 0.05 was considered statistically significant. Statistical analysis was performed using GraphPad Prism 6.0 software (GraphPad Software). Samples sizes were estimated based on previous studies. All samples represent biological replicates. Representative western blots and EMSAs were repeated twice. For mouse studies, data were excluded if values could not be obtained from the sample or equipment malfunction prevented obtaining data. Randomization was not used. For physiology experiments, the investigator was blinded to the group being analyzed.

**Data availability.** The data that support the finding of this study are available from the corresponding author on request. Microarray and ChIP-seq data have been deposited into the GEO database under accession code GSE114026. SubSeries: GSE114023 and GSE114025.

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

## Acknowledgements

We are grateful to J.A. Rafael-Fortney, M.R. Parthun, and past and present members of the Guttridge lab for their helpful discussions throughout this study. We thank B.P. Ashburner for kindly sharing reagents. We very much appreciate J. Brice and D. Bryant in the Ohio State University Solid Tumor Biology Program histology core for their patience and assistance preparing whole heart sections, and S. Cole and B. Kemmenoe in the OSU Campus Microscopy and Imaging Facility for their assistance in capturing images. Support for this work was provided by grants from the National Institutes of Health: U01 NS058451 (D.C.G. and P.M.L.J.), F32 HL099145 (J.M. Peterson), R01 AR044719 (S.M.S.), as well as the Muscular Dystrophy Association (J.M. Peterson), and Parent Project Muscular Dystrophy (D.C.G.).

## Author contributions

J.M. Peterson designed the study, collected, and analyzed the data, and wrote the manuscript. D.J.W. provided experimental expertize and guidance throughout the study. V.S., S.R.R., B.D.C., and S.C.L. performed physiology measurements. N.B., J.S., and J.-M. G. provided experimental expertize for various aspects of the study. L.L. performed ChIP sequencing. N.M.R., C.E.G., and J.M. Petrosino assisted with data collection. P.L. performed co-immunoprecipitations, S.L. assisted with microarray analysis. H.W. provided expertize with and performed ChIP sequencing. P.M.L.J., J.P.D., and M.T.Z. designed and assisted with analysis and interpretation of physiology data. S.M.S. assisted in bioinformatics pipeline development and bioinformatics analysis and provided expertize in ChIP assays. D.C.G. designed and mentored the study and assisted in preparation of the manuscript. All authors approved the final manuscript version.

## Additional information

**Competing interests:** The authors declare no competing interests.

