## [Peer Review File · Nature Communications]

Reviewers' comments:

Reviewer #1 (Remarks to the Author):

This is a very interesting manuscript reporting on the contribution of activated NF- κ B signaling to the cardiac phenotype in DMD. The authors show that genetic ablation of NF- κ B in cardiomyocytes of mdx mice, by conditional deletion of IKK β , can rescue cardiac abnormalities through the up-regulation of calcium handling genes. This finding correlates with global enrichment of transcription-permissive histone marks, such as H3K27 acetylation, and depletion of transcriptional repressors, such as the "architectural protein" CTCF, and components of the deacetylase co-repressory complex SIN3A and HDAC1. These findings indicate novel targets for interventions aimed at alleviating cardiac pathology in DMD (an issue of extreme importance in the field) and at the same time suggest an unconventional and previously unrecognized repressive activity of NF- κ B on calcium homeostasis in DMD hearts.

In general, the experiments are well conducted and data support the author's conclusions. The information is of medical as well as basic transcription interest, and should be definitely made available to the scientific community.

There are some critical issues that the authors should address to further improve the quality of manuscript before it is published.

- 1) Most of the conclusions on the connection between NF κ B-mediated gene repression, histone acetylation and co-repressors (Sin 3 and HDAC1) are mainly based on association data. More conclusive evidence should be provided by the authors. The authors can use the available tools to address this issue. I suggest that the authors expose cardiomyocytes, isolated from WT or conditional NF κ B mutants, to the NF κ B activator TNF- α alone or in combination with the HDAC inhibitor TSA, and check the expression of target genes (Slc8a1, Rcan1, Cacna1h and Camk4). If NF κ B cooperates with HDAC1 in repressing these genes, then the proposed experiment should show that TNF α and HDAC antagonize each other in WT cells, but not in NF κ B-deficient cells.
- 2) Most of the experiments are performed in mdx mice at 6 or 7 months of age. The rationale for the selection of this time point is unclear and should be explained
- 3) The NF κ B-mediated reduction in histone acetylation was also observed in MEFs. The authors suggest that this is an evidence for a common mechanism conserved among cell types. However, NF κ B-mediated repression of calcium handling genes seems to be a specific feature of cardiomyocytes. The authors should verify this by comparison of effect of NF κ B deficiency in cardiac vs skeletal muscle cells.
- 4) Likewise, it is unclear whether aberrant NF κ B-mediated gene repression is a specific feature of DMD muscles or is also observed in cardiomyocytes of normal mice. This should also be verified by comparing NF κ B deficiency in cardiomyocytes from normal vs dystrophic muscles.
- 5) The cooperation between NF κ B and HDAC in repressing gene expression by reducing H3K27Ac is one of the most exciting data of this manuscript and is rightly mentioned in the abstract. However, the mechanism beyond this cooperation is not discussed. Likewise, the potential implication of this finding on the beneficial effect of HDAC inhibitors in mdx mice should also be discussed.

Reviewer #2 (Remarks to the Author):

From an NF- κ B point of view, this is a very interesting paper describing a novel function for NF- κ B signaling in disease.

Some points for the authors to address:

1. Please label figure 1A (left panel) more clearly. Instead of NF- κ B please indicate which subunit is being detected.
2. Were any of the experiments using NBD peptide confirmed with the IKK β inhibitor Compound

A?

3. Have any non-canonical NF- κ B members been investigated in the mdx control/IKK β KO hearts to determine if this pathway might be partly responsible for the observed phenotypes? Please discuss.
4. Did any of downregulated genes in the mdx/HRT Δ IKK β cardiomyocytes stand out as potentially contributing to the improved phenotype?
5. Is it possible to confirm the experiment in figure 3A in cardiomyocytes? I realize that MEFs must be easier to work with, but this could provide more impact as you then show acetylation differences between hearts from your animal models.

Reviewer #3 (Remarks to the Author):

In this article entitled "NF- κ B promotes cardiac dysfunction through global epigenetic repression of calcium homeostatic genes in a model of Duchenne muscular dystrophy", Peterson et. al. report that genetic ablation of NF- κ B signaling in cardiomyocytes rescue the cardiac dystrophic phenotype in mdx mice. The authors suggest that this improvement is associated with an upregulation of calcium handling genes. They propose that in this context, NF- κ B instead of acting as a transcription factor, it causes a global repression of calcium homeostatic genes.

Specific Comments:

1) Though the authors have slightly touched on early and late time point of signaling and hence cardiac dysfunction associated with it in mdx mice, analyses are done at different time point making drawing conclusion a bit difficult. For example, cardiac analyses for NF- κ B and p65/p50 expressions are done at 1 year and 3 months of age while the cardiac phenotyping was performed at 13-14 months of age and fibrosis analysis of the cardiac tissue was done at 6 months of age.

2) Figure 1C shows increased phospho-p65 in cardiac sections, claiming that this is specific to cardiomyocytes. Since this is an important point, given the pleiotropic role of NF- κ B in many cells types and the presence of inflammation in diseased hearts, it is best to perform double staining of phosphor-p65 with a cardiac specific marker (such as cardiac Troponin T or α -actinin) that is known to work well on cardiac sections.

3) A few of the graphs are missing WT control bars making it difficult to conclude whether the readout is indeed an issue in mdx mice. For example, Figures 1J, 1L, 1M, 2D, supplemental Figure 2A, and Supplemental Figure 3A. Especially, given the role of NF- κ B in modulating AngII and isoproterenol induced cardiac hypertrophy in vivo (Freund et. al., 2005, Circulation; Maier et. al., 2012, PNAs), the authors need to include both WT and HRT Δ IKK2 β controls in these experiments.

4) Prolonged activation of NF- κ B in the heart has detrimental effects (Kraut et. al., 2015 Plos One; Hamid et. al., 2011, Cardiovascular Research), while cardiac specific deletion of NEMO (IKK γ) subunit has been previously shown to cause dilated cardiomyopathy (Kratsios et. al., 2010 Circ Res). Given the multiple facets of NF- κ B in the heart, with reports showing either maladaptive and cardioprotective role for NF- κ B, the authors cannot simply ignore all the previous literature but instead they need to discuss their findings in light of these previous findings. For example, it is possible that the actions of the NEMO protein itself might be multifaceted (i.e. canonical versus non canonical roles, potential posttranslational modifications, linear polyubiquitin chains). More importantly, they need to include the HRT Δ IKK2 β mouse as an important control in their experiments.

5) The authors should comment whether the improvement of force (Figure 1J) after NDB treatment was accompanied by other histological (i.e. fibrosis) or signaling changes (i.e. calcium homeostatic genes). This is an important point if this might serve as a potential treatment and the authors need to expand on this findings including experiments that support the signaling changes

accompany such treatment.

6) As it is well established that hearts from mdx mice suffer from mitochondrial irregularities leading to dysfunctional oxygen production and ATP formation, one would hesitate on the optimistic result of having a heart that responds to Dobutamine to a higher level (and hence higher myocardial oxygen demand). Please expand on this.

7) As closed chest P-V loop analysis is performed and the authors are suggesting the presence of diastolic dysfunction with preserved systolic function, it is best to include cardiac systolic parameters including but not limited to contractility index, systolic pressure and systolic volume. When performing closed chest P-V loop analysis diastolic as well as systolic parameter values are available. The authors should include whole heart histology such as H&E staining as well as P-V loop measurements for all three groups as they provide significant amount of information.

8) Some of the parameters do not match with previously observed cardiac function analyses such as ejection fraction, tau value, diastolic functions, (i.e. Wasala et. al., 2017, JMCC). The authors need to discuss why these parameters are different in their hands compared with these studies.

9) Please include cardiac mass measurements as this helps identifying the state of the heart; based on provided data the heart is not exhibiting dilated cardiomyopathy however it still remains unknown if the heart is hypertrophic or exhibiting stiffness at 13-14 m of age. It is well established that in DMD patients early diastolic dysfunction and focal fibrosis proceed to dilated cardiomyopathy.

10) Could the fibrosis come about later on? Though it is well established that DMD patients suffer from fibrotic cardiac tissues early on during the progression of the disease, the authors' H&E staining at 1year of age (Figure 1C) shows the elevated extracellular matrix area of the myocardium possibly suggestive of elevated fibrosis; then it would be best to also analyze cardiac tissues at 1year of age this confirms the used model.

11) The calcium amplitude results (Figure 2B) are showing the opposite of previously published data (including some of the references included in this manuscript). This could be due to the experimental approaches i.e. instead of evaluating individual calcium sparks the authors are using mean intensity of cells which may blind or average out the actual calcium at the time of release. Please expand on the obtained results while covering available data in the literature. It is suggested to quantify available cytosolic calcium levels in isolated cells in order to answer the discrepancy. Further, the lower calcium amplitude possibly suggests lower levels of available calcium which would raise the question that how would the mdx mouse used here would not have systolic dysfunction if low levels of calcium are available.

12) In Figure 5I, the authors claim that "the overall binding density of SIN3A and HDAC1 was greater surrounding p65 peaks in genes that are upregulated in IKK β knockout hearts". However, they should present some data from the analysis of IKK β knockout hearts. Moreover, p65 has been shown to interact with HDAC1 and HDAC2 to negatively regulate gene expression (Ashburner et. al., 2001, MCB). Do the authors imply that p65 recruits HDAC1 or there is something else going on? A schematic representation of their hypothesis would be helpful.

Minor Comments:

1) Please only include data from available databases vs. discrete ones.

2) There are large variations among some of the results within the same figure: i.e.. Figure 2A.

3) Figure 4A look identical between the 2 groups. The authors need to present this message in a different way or at least, they should include numbers (%) for the different categories so the changes in the proportion of bound regions within Intergenic and Distal intergenic regions would

be appreciated.

4) There are a few "data not shown"; Please include all data that support the work. If some of the "data not shown" can not be presented, please draw conclusions based on presented data and it is suggested to exclude unpublished data or data that do not meet the quality of publication.

5) If Slc8a1 effect is such a generalized phenomenon, perhaps the authors can analyze the skeletal muscle of their previously published mdx/IKK2muscle specific KO. This will strengthen their conclusions.

6) Please expand introduction to include previously publications using mdx mice.

7) In the discussion, please mainly cover studies on DMD rather than other cardiovascular diseases.

Reviewer #4 (Remarks to the Author):

Peterson et al. are investigating the pathogenic mechanism by which NF- κ B mediates cardiac dysfunction in DMD. Their focus is on the p65/RelA component and propose that it functions in aberrant global epigenetic repression and focus on a subset of genes involved in calcium regulation. This study revealed some interesting and unexpected results that elucidate broadly how aberrant NF- κ B activation in DMD leads to cardiac pathology and additionally shows a potential pathway to therapeutic development, however, there are some concerns detailed below:

Title is misleading, this is more accurately "...through repression of calcium homeostatic genes ..." as opposed to "global epigenetic repression" as they show no evidence for any true epigenetic regulation (DNA methylation or polycomb/trithorax involvement). Similarly, there is no data presented on chromatin conformation, just chromatin content and only changes in active chromatin marks, no attempts to assay repressive marks (e.g. DNA methylation or H3K9me3 or H3K27me). Histone acetylation is typically associated with active chromatin but is often just an interaction face for transcriptional activators. Do DNaseI hypersensitive sites ever change in absence of NF- κ B? 3C interactions?

Line 196: Should be noted that p65(RelA) is reported to directly interact with HDAC1 and also function as a repressor in certain contexts so this is not completely unexpected.

Line 201: Why a surprise? Typically, you are activating gene expression (e.g. pro-inflammatory genes) with NF- κ B activation but also affecting a major gene regulatory network; p65 can be a repressor interacting with HDAC1, thus they are removing a known repressor. Also, p50 homodimers are repressive, what happens to their distribution and formation; how does removal of p65 affect the rest of the NF- κ B dimer and heterodimer formations and targets?

Have an issue that NF- κ B (p65/p50 heterodimer) is a global repressor in mdx; don't think they have made the case. It would be very helpful if expression was put into the context of WT hearts.

Supp Fig 3C, only a moderate 2-fold enrichment for H3K27Ac at Slc8a1 regulatory region? Is this just loss of repression and not really activation?

Are any of these genes differentially methylated in WT vs mdx or +/- p65? It would be somewhat surprising for chronically (?) silenced genes, such as these appear in mdx, with CpG islands to not be methylated, but this is not addressed at any level. The general lack of information on the repression mechanism (other than HDA1/2 which is in many, many chromatin complexes) for a manuscript with "repression" in the title is frustrating.

Authors state that they investigated several repressor proteins and only SIN3A and HDAC1 were present. Should list which ones and include data as supplement as it will strengthen the specificity of the assays. Overall, I am cautious with data that shows similar changes globally for a very specific phenotype.

Comparisons throughout are all in mdx with or without p65, however, it is important to know the comparison with WT hearts. Gene expression levels change and histone marks change but are they back to normal healthy levels and patterns?

It is surprising that DNA methylation was not really investigated since the Slc8a1 regulatory region is a CpG island and they over riding claim is an epigenetic mechanism. Regardless, the data supports typical transcription factor-mediated repression/activation at these genes and not true epigenetic regulation. The aza-C experiment was weak, particularly since AzaC can in instances activate gene expression without affecting DNA methylation so it is not really clear the control for demethylation even worked.

Statistics for the epigenetic studies are appropriate

We would like to thank each of the reviewers for their thoughtful and constructive comments. We were pleased that you found interest in our study and remarked on the novelty of our findings with regards to the function of NF- κ B in regulating cardiomyopathy in Duchenne muscular dystrophy. Your input has been extremely helpful in improving and clarifying our manuscript. Although there was general acceptance of our findings, your critiques required several of our figures to be substantially revised, specifically regarding the addition of wild-type mice, which needed to be aged matched and thus added to the time of our revision. In addition, numerous *in vivo* and *in vitro* experiments were also modified or added to address your concerns. In all, this meant the revision or addition of 25 figure panels and two new tables. We acknowledge that these new data have substantially improved the quality of our study, which we believe you will appreciate as well. These changes are detailed below in our point-by-point responses. We again thank you for your comments and time spent during the review process.

Specific Responses to Reviewer #1:

This is a very interesting manuscript reporting on the contribution of activated NF- κ B signaling to the cardiac phenotype in DMD. The authors show that genetic ablation of NF- κ B in cardiomyocytes of mdx mice, by conditional deletion of IKKbeta, can rescue cardiac abnormalities through the up-regulation of calcium handling genes. This finding correlates with global enrichment of transcription-permissive histone marks, such as H3K27 acetylation, and depletion of transcriptional repressors, such as the “architectural protein” CTCF, and components of the deacetylase co-repressory complex SIN3A and HDAC1. These findings indicate novel targets for interventions aimed at alleviating cardiac pathology in DMD (an issue of extreme importance in the field) and at the same time suggest an unconventional and previously unrecognized repressive activity of NF- κ B on calcium homeostasis in DMD hearts.

In general, the experiments are well conducted and data support the author’s conclusions. The information is of medical as well as basic transcription interest, and should be definitely made available to the scientific community.

Answer: We thank the reviewer for recognizing the novelty of our work and for the general support of our findings.

There are some critical issues that the authors should address to further improve the quality of manuscript before it is published.

1) Most of the conclusions on the connection between NF κ B-mediated gene repression, histone acetylation and co-repressors (Sin 3 and HDAC1) are mainly based on association data. More conclusive evidence should be provided by the authors. The authors can use the available tools to address this issue. I suggest that the authors expose cardiomyocytes, isolated from WT or conditional NF κ B mutants, to the NF κ B activator TNF-alpha alone or in combination with the HDAC inhibitor TSA, and check the expression of target genes (Slc8a1, Rcan1, Cacna1h and Camk4). If NF κ B cooperates with HDAC1 in repressing these genes, then the proposed experiment should show that TNF alpha and HDAC antagonize each other in WT cells, but not in NF κ B-deficient cells.

Answer: We understand the reviewer’s desire to have more mechanistic evidence of NF- κ B-mediated gene repression. Regarding the suggested experiment, while we appreciate the concept of the experiment proposed by the reviewer with NF- κ B mutant cardiomyocytes, unfortunately it presents a significant technical challenge for two reasons. First, mouse cardiomyocytes are viable for less than 1 day and therefore treatment with TSA and TNF would not be possible. This is why with the exception of experiments that can be performed immediately post-isolation, longer culturing of cardiomyocytes is performed in rat, rabbit, or human cardiomyocytes, which can survive longer periods of time. Second, cells deficient in NF- κ B signaling undergo apoptosis when exposed to TNF, and therefore any results obtained after treating NF- κ B-deficient cells with

TNF could be due to cell death signaling rather than direct NF- κ B-mediated processes. With that said, we agree with the concept of the experiment mentioned by the reviewer and have addressed the question using wt cells. We treated HL-1 cardiomyocytes with TSA or TSA + TNF. As the reviewer suggested, since NF- κ B cooperates with SIN3A and HDAC1 in repressing genes, data show that TNF and HDAC do indeed antagonize each other. These data are presented in revised Fig 5I.

In addition, we have attempted to provide more evidence that NF- κ B is mediating gene repression, by showing for the first time that p65 interacts with SIN3A (revised Supplementary Figure 5E which appears in the Discussion section page 20). This result also bridges a link to HDAC1 since both p65 and SIN3A have previously been shown to interact with HDAC1.

2) *Most of the experiments are performed in mdx mice at 6 or 7 months of age. The rationale for the selection of this time point is unclear and should be explained.*

Answer: Thank you for pointing this out. In brief, we now provide a rationale for choosing a timeline for cardiomyopathy in *mdx* mice, which is based on published studies and explained in the first paragraph of the results section. Additionally, each time we present an experiment we now reinforce our justification for using a particular time point based on the time line introduced in the results section.

3) *The NF κ B-mediated reduction in histone acetylation was also observed in MEFs. The authors suggest that this is an evidence for a common mechanism conserved among cell types. However, NF κ B-mediated repression of calcium handling genes seems to be a specific feature of cardiomyocytes. The authors should verify this by comparison of effect of NF κ B deficiency in cardiac vs skeletal muscle cells.*

Answer: We have added CHIP data for H3K27ac on C2C12 skeletal myoblasts that are wt or deficient for NF- κ B (C2C12SR). Our results show that NF- κ B-mediated reduction of histone acetylation also occurs in skeletal muscle cells (revised Fig. 3H).

4) *Likewise, it is unclear whether aberrant NF κ B-mediated gene repression is a specific feature of DMD muscles or is also observed in cardiomyocytes of normal mice. This should also be verified by comparing NF κ B deficiency in cardiomyocytes from normal vs dystrophic muscles.*

Answer:

We recognize that this is an interesting question that we have considered, but we also see this as a broader question, which would change the focus of our paper.

To address it would require a substantial increase in mouse numbers throughout the entire study. So as to not ignore the question, we examined *Slc8a1* gene expression in a small set of wt hearts containing or deficient for NF- κ B. No differences were observed between the two groups at 3 months, but a non-significant increase started to become visible at 8 months of age (Fig.1). More mice and perhaps additional time points will be needed to determine if this difference will become significant. In summary, whether NF- κ B controls the homeostatic function of cardiac muscle through a similar mechanism that we described in our current work will be investigated in a later, more comprehensive study.

Figure 1. Relative *Slc8a1* gene expression determined from whole hearts. wt indicates no dystrophin mutation. Mice contained either intact (IKK β / β) or cardiomyocyte-specific deletion (HRT Δ IKK β) for NF- κ B.

5) *The cooperation between NFκB and HDAC in repressing gene expression by reducing H3K27Ac is one of the most exciting data of this manuscript and is rightly mentioned in the abstract. However, the mechanism beyond this cooperation is not discussed. Likewise, the potential implication of this finding on the beneficial effect of HDAC inhibitors in mdx mice should also be discussed.*

Answer: To address this point, we have now added further discussion in our paper commenting both on the mechanism by which NF-κB is able to reduce H3K27Ac and the potential beneficial effects of administering HDAC inhibitors for DMD therapy, in the context of what has already been proposed in the literature for dystrophic skeletal muscles (see revised page 20). We would like to make one clarification regarding our previous interpretation of how activation of NF-κB in dystrophic hearts leads to the reduction of calcium handling genes by remodeling chromatin through inhibition of H3K27ac. As a result of new data we have added comparing wt versus *mdx* hearts, we now report in our manuscript that expression of the calcium handling genes is not reduced in *mdx* hearts even though H3K27ac is clearly depleted. Only upon inhibiting NF-κB in *mdx* hearts do we see the increase in expression of calcium genes concurrently with the restoration of H3K27ac and the concomitant decrease in CTCF, HDAC1, and Sin3a chromatin association. We reason, as explained in the Discussion section of our manuscript, that in a dystrophic condition the heart is likely trying to compensate for its functional deficiency by attempting to stimulate the transcription of calcium handling genes. However, this compensatory reaction is negated through the activation of NF-κB and association with CTCF, HDAC1, and Sin3a, which reduces H3K27ac and renders chromatin less active.

Specific Responses to Reviewer #2 :

From an NF-κB point of view, this is a very interesting paper describing a novel function for NF-κB signaling in disease.

Answer: We thank the reviewer for recognizing the novelty of our findings.

Some points for the authors to address:

1. *Please label figure 1A (left panel) more clearly. Instead of NF-κB please indicate which subunit is being detected.*

Answer: The EMSA we showed only indicates that NF-κB binding is occurring, but does not designate which exact dimer complexes are involved. From the supershifts we performed, we can presume that the top band in the complex contains the p65 subunit and the larger bottom band contains p50, but in our hands performing supershift analyses with commercially available antibodies against the other subunits (p52, c-Rel, RelB) has not been conclusive. This is why we prefer to designate the binding complexes in Fig. 1A as simply NF-κB.

2. *Were any of the experiments using NBD peptide confirmed with the IKKβ inhibitor Compound A?*

Answer: We agree that this would be a useful experiment, but unfortunately Compound A has not been sufficiently investigated in *mdx* mice. Because of the wealth of pre-clinical data we have accumulated with the NBD peptide in both mouse and dog models of DMD, we focused on this inhibitor for our studies.

3. *Have any non-canonical NF-κB members been investigated in the mdx control/IKKβ KO hearts to determine if this pathway might be partly responsible for the observed phenotypes? Please discuss.*

Answer: We have now included a western blot for p100/p52 processing, using spleen as a positive control. These data in revised Supplementary Fig. 1E show that processing of p100 to p52 does not change in normal versus *mdx* hearts. These data further suggest that activation of NF-κB in the dystrophic heart likely reflects the classical signaling pathway.

4. Did any of downregulated genes in the *mdx/HRTΔIKKβ* cardiomyocytes stand out as potentially contributing to the improved phenotype?

Answer: No, the downregulated genes did not stand out as potentially contributing to the improved phenotype. We were initially surprised that downregulated genes did not provide us a direction to explore, which is why we decided to take a broader view on the upregulated genes.

5. Is it possible to confirm the experiment in figure 3A in cardiomyocytes? I realize that MEFs must be easier to work with, but this could provide more impact as you then show acetylation differences between hearts from your animal models.

Answer: We agree with the reviewer that this experiment strengthens the study and have now performed the TSA experiment on the HL-1 cardiomyocyte cell line. The results, which are presented in revised Fig. 3C show that *Slc8a1* gene expression is increased when HL-1 cells are exposed to TSA, which is consistent to the data obtained in MEFs. In response to a suggestion from Reviewer 1 point 1, we have additionally analyzed gene expression of all four calcium handling genes in HL-1 cardiomyocytes (Revised Fig. 5I) and found that TSA administration stimulated expression of these target genes.

Specific Responses to Reviewer #3:

In this article entitled “NF-κB promotes cardiac dysfunction through global epigenetic repression of calcium homeostatic genes in a model of Duchenne muscular dystrophy”, Peterson et. al. report that genetic ablation of NF-κB signaling in cardiomyocytes rescue the cardiac dystrophic phenotype in mdx mice. The authors suggest that this improvement is associated with an upregulation of calcium handling genes. They propose that in this context, NF-κB instead of acting as a transcription factor, it causes a global repression of calcium homeostatic genes.

Specific Comments:

1) *Though the authors have slightly touched on early and late time point of signaling and hence cardiac dysfunction associated with it in mdx mice, analyses are done at different time point making drawing conclusion a bit difficult. For example, cardiac analyses for NF-κB and p65/p50 expressions are done at 1 year and 3 months of age while the cardiac phenotyping was performed at 13-14 months of age and fibrosis analysis of the cardiac tissue was done at 6 months of age.*

Answer: This was a point that was also raised by Reviewer 1. We agree that this was confusing, so we now provide rationale for choosing particular time points. In brief, we now provide a rationale for choosing a timeline for cardiomyopathy in *mdx* mice, which is based on published studies and explained in the first paragraph of the results section. Additionally, we have eliminated the fibrosis data from 6 months and replaced it with 1-year data to be consistent with our *in vivo* functional experiments (Supplementary Figs 2B and 2C).

2) *Figure 1C shows increased phospho-p65 in cardiac sections, claiming that this is specific to cardiomyocytes. Since this is an important point, given the pleiotropic role of NF-κB in many cells types and the presence of inflammation in diseased hearts, it is best to perform double staining of phosphor-p65 with a cardiac specific marker (such as cardiac Troponin T or a-actinin) that is known to work well on cardiac sections.*

Answer: To address the reviewer’s point, we have enhanced our original images to show that localization of phospho-p65 occurs in striated cells, which strongly suggest that these are cardiomyocytes. As the reviewer recommended, we have also performed double staining with alpha-sarcomeric actin to confirm that phospho-p65 staining is cardiomyocyte-specific. These new data appear in revised Supplementary Figure 1A.

3) *A few of the graphs are missing WT control bars making it difficult to conclude whether the readout is indeed an issue in mdx mice. For example, Figures 1J, 1L, 1M, 2D, supplementary Figure 2A, and Supplementary Figure 3A. Especially, given the role of NF- κ B in modulating AngII and isoproterenol induced cardiac hypertrophy in vivo (Freund et. al., 2005, Circulation; Maier et. al., 2012, PNAs), the authors need to include both WT and HRT Δ IKK2 β controls in these experiments.*

Answer: We thank the reviewer for this suggestion, particularly as it pertains to adding wt heart expression data, as this has strengthened our paper and clarified the role of NF- κ B in dystrophic hearts. We have now added wt data in 14 figure panels, which include: Revised Figs. 1J, 1K, 1L, 1M, 2D, 3J, 4F, and Revised Supplementary Figs. 1B, 1C, 1D 1E, 2B (formerly Supplementary Fig. 2A), 2C (formerly Supplementary Fig. 2B), and 3A. We have also added wt results in Table 1 and Supplementary Table 1.

4) *Prolonged activation of NF- κ B in the heart has detrimental effects (Kraut et. al., 2015 Plos One; Hamid et. al., 2011, Cardiovascular Research), while cardiac specific deletion of NEMO (IKK γ) subunit has been previously shown to cause dilated cardiomyopathy (Kratsios et. al., 2010 Circ Res). Given the multiple facets of NF- κ B in the heart, with reports showing either maladaptive and cardioprotective role for NF- κ B, the authors cannot simply ignore all the previous literature but instead they need to discuss their findings in light of these previous findings. For example, it is possible that the actions of the NEMO protein itself might be multifaceted (i.e. canonical versus non canonical roles, potential posttranslational modifications, linear polyubiquitin chains). More importantly, they need to include the HRT Δ IKK2 β mouse as an important control in their experiments.*

Answer: We agree with the reviewer that NF- κ B has been shown to perform multiple functions in the heart, and we have done our best in our revised manuscript to discuss these various functions as it relates to our current findings in both the Introduction and Discussion sections of our paper. With regards to the addition of data from mice with hearts deficient in NF- κ B but wt for the dystrophin mutation, please refer to our response to Reviewer 1, point 4.

5) *The authors should comment whether the improvement of force (Figure 1J) after NBD treatment was accompanied by other histological (i.e. fibrosis) or signaling changes (i.e. calcium homeostatic genes). This is an important point if this might serve as a potential treatment and the authors need to expand on this findings including experiments that support the signaling changes accompany such treatment.*

Answer: NBD functional data in this manuscript was added as a bridge between our previous publication that showed functional cardiac improvements when dko mice were treated with NBD (Delfin et al 2011), and this manuscript in which we are showing a mechanism behind this rescue. Any data obtained from systemic administration of an inhibitor such as the reviewer suggests cannot directly be attributed to inhibition in a particular cell type, as can be mechanistically dissecting by using a genetic ablation of cardiomyocyte-specific IKK β . Therefore, while we appreciate the reviewer's comment, we needed to prioritize where the critical places were in this manuscript for using the aged mice we had available to perform all the necessary revisions. For this reason, we hope the reviewer understands why we are reluctant to perform additional NBD experiments.

6) *As it is well established that hearts from mdx mice suffer from mitochondrial irregularities leading to dysfunctional oxygen production and ATP formation, one would hesitate on the optimistic result of having a heart that responds to Dobutamine to a higher level (and hence higher myocardial oxygen demand). Please expand on this.*

Answer: We apologize if this was not clear in the original text of our paper. We have now clarified this point in our revised text to emphasize that failure to respond to beta agonists such as Isoproterenol and Dobutamine is indicative of a failing heart and not a direct reflection of whether or not it is good to challenge a diseased

heart (see revised page 9). Normalized response to sympathetic stimulation is therefore used as a readout for how sick the heart is. Thus, our results showing that *mdx* hearts deficient in NF- κ B generated greater force in response to Dobutamine is an indicator that NF- κ B promotes cardiac dysfunction in *mdx* hearts and that inhibiting it results in a better performing heart.

7) *As closed chest P-V loop analysis is performed and the authors are suggesting the presence of diastolic dysfunction with preserved systolic function, it is best to include cardiac systolic parameters including but not limited to contractility index, systolic pressure and systolic volume. When performing closed chest P-V loop analysis diastolic as well as systolic parameter values are available. The authors should include whole heart histology such as H&E staining as well as P-V loop measurements for all three groups as they provide significant amount of information.*

Answer: We agree with the reviewer that inclusion of the suggested data will be useful to readers. We now show whole heart histology (revised Supplementary Fig. 1B) and have added two tables (Table 1 and Supplementary Table 1) that include complete echocardiogram and PV loop measurements.

8) *Some of the parameters do not match with previously observed cardiac function analyses such as ejection fraction, tau value, diastolic functions, (i.e. Wasala et. al., 2017, JMCC). The authors need to discuss why these parameters are different in their hands compared with these studies.*

Answer: It is difficult to compare measurements from mice with different dystrophin mutations. *mdx*^{2cv}, *mdx*^{3cv}, *mdx*^{4cv}, and *mdx*^{5cv} are all used in the literature and are known to have differences that will effect function such as containing less revertant fibers. The particular study the reviewer mentioned did not use the traditional *mdx* mouse as we used in our study, but the *mdx*^{3cv} and *mdx*^{4cv} strains. Therefore, we believe our results do not differ from those currently in the literature using the traditional *mdx* mouse model, only those with other mutations and phenotypes.

9) *Please include cardiac mass measurements as this helps identifying the state of the heart; based on provided data the heart is not exhibiting dilated cardiomyopathy however it still remains unknown if the heart is hypertrophic or exhibiting stiffness at 13-14 m of age. It is well established that in DMD patients early diastolic dysfunction and focal fibrosis proceed to dilated cardiomyopathy.*

Answer: Cardiac mass measurements have now been added and are included in a new table (Table 1).

10) *Could the fibrosis come about later on? Though it is well established that DMD patients suffer from fibrotic cardiac tissues early on during the progression of the disease, the authors' H&E staining at 1year of age (Figure 1C) shows the elevated extracellular matrix area of the myocardium possibly suggestive of elevated fibrosis; then it would be best to also analyze cardiac tissues at 1year of age this confirms the used model.*

Answer: Yes, we agree with the reviewer that 6 months might have been too early to determine fibrosis, and our analysis was not performed at time points consistent with the majority of the force measurements we made at 13-14 months of age. In our revised paper we have now removed the fibrosis data at 6 months and replaced it with new fibrosis measurements made at 12 months in wt, *mdx* and NF- κ B-deficient *mdx* hearts. The new data in revised Supplementary Fig. 2C are consistent with the earlier time point in that we are unable to find a difference in fibrosis between *mdx* hearts with and without NF- κ B signaling. Images representative of the quantitated hearts are also included in revised Supplementary Fig. 2B.

11) *The calcium amplitude results (Figure 2B) are showing the opposite of previously published data (including some of the references included in this manuscript). This could be due to the experimental approaches i.e. instead of evaluating individual calcium sparks the authors are using mean intensity of cells which may blind or average out the actual calcium at the time of release. Please expand on the obtained*

results while covering available data in the literature. It is suggested to quantify available cytosolic calcium levels in isolated cells in order to answer the discrepancy. Further, the lower calcium amplitude possibly suggests lower levels of available calcium which would raise the question that how would the mdx mouse used here would not have systolic dysfunction if low levels of calcium are available.

Answer: The calcium transient amplitude results are consistent with previously published data. We apologize for the confusion if we gave the reviewer the impression that we were measuring calcium sparks rather than calcium transient amplitudes. We have revised the text to more clearly indicate these are calcium amplitudes rather than calcium sparks (page 11).

12) In Figure 5I, the authors claim that “the overall binding density of SIN3A and HDAC1 was greater surrounding p65 peaks in genes that are upregulated in IKK β knockout hearts”. However, they should present some data from the analysis of IKK β knockout hearts. Moreover, p65 has been shown to interact with HDAC1 and HDAC2 to negatively regulate gene expression (Ashburner et. al., 2001, MCB). Do the authors imply that p65 recruits HDAC1 or there is something else going on? A schematic representation of their hypothesis would be helpful.

Answer: We apologize for the confusion. The point of this analysis is to show that the overall binding density of SIN3A and HDAC1 is greater surrounding p65 peaks. Since our CHIP-seqs from IKK β knockout hearts were performed with a H3K27ac and not a p65 antibody, this would not illustrate our point. We can see how this figure can be confusing and to make this clearer to the reader we have now revised the text to more explicitly state this. We have also added a paragraph in the discussion section to discuss how we think p65-HDAC1-SIN3A complex together (page 20).

Minor Comments:

1) Please only include data from available databases vs. discrete ones.

Answer: The data we used were only obtained from publically available databases. We have ensured that GSM numbers for each of the publically available datasets have been included in the Methods section of our manuscript (see revised page 29).

2) There are large variations among some of the results within the same figure: i.e.. Figure 2A.

Answer: Yes, we acknowledge that there is a large mouse to mouse variability in the *mdx* mouse, however the majority of our measures show statistical differences, demonstrating that even though large variations occur in these mice, our results are quite strong.

3) Figure 4A look identical between the 2 groups. The authors need to present this message in a different way or at least, they should include numbers (%) for the different categories so the changes in the proportion of bound regions within Intergenic and Distal intergenic regions would be appreciated.

Answer: The illustration in Fig. 4A is quite standard in the way such CHIP-seq data are presented from bioinformatic analysis. This analysis is genome wide showing the global H3K27ac regions. Although NF- κ B deficient *mdx* hearts contained approximately 10,000 more H3K27ac regions, globally, the fact that these panels look identical illustrates our point perfectly. H3K27ac should be globally bound to enhancer regions in both CHIP-seqs, the difference is which genes H3K27ac is bound to, not where within the genome it binds. We go on to show that there are differences in enrichment at particular genes. However, we understand why this posed confusion, and have now attempted to clarify this point in the revised text (page 14).

4) There are a few "data not shown"; Please include all data that support the work. If some of the "data not shown" can not be presented, please draw conclusions based on presented data and it is suggested to exclude unpublished data or data that do not meet the quality of publication.

Answer: All places where we had indicated “data not shown” in our previous manuscript has been changed. We now either include the data or the have removed the statements entirely.

5) *If Slc8a1 effect is such a generalized phenomenon, perhaps the authors can analyze the skeletal muscle of their previously published mdx/IKK2 muscle specific KO. This will strengthen their conclusions.*

Answer: We believe our manuscript provides findings that are quite novel with respect to how NF- κ B negatively regulates calcium genes and cardiomyocyte dysfunction without affecting the histopathology of *mdx* hearts (clearly different from how NF- κ B functions in *mdx* skeletal muscle). We value the comment of the reviewer, but we also believe performing a separate analysis in skeletal muscle at this time would detract from the main message of our study.

6) *Please expand introduction to include previously publications using mdx mice.*

Answer: We have tried to the best of our ability to now include all relevant citations throughout the revised text. The only publication that we are aware of on NF- κ B in *mdx* hearts was performed by our group and is included in the Introduction section.

7) *In the discussion, please mainly cover studies on DMD rather than other cardiovascular diseases.*

Answer: The decision to expand to other cardiovascular diseases in our Discussion section was done with the intent to relate our relevant studies back to our results. Additionally, we wrote this section again with the intent to appeal to the wide audience Nature Communications attracts and not just for DMD scientists.

Specific Responses to Reviewer #4:

Peterson et al. are investigating the pathogenic mechanism by which NF- κ B mediates cardiac dysfunction in DMD. Their focus is on the p65/RelA component and propose that it functions in aberrant global epigenetic repression and focus on a subset of genes involved in calcium regulation. This study revealed some interesting and unexpected results that elucidate broadly how aberrant NF- κ B activation in DMD leads to cardiac pathology and additionally shows a potential pathway to therapeutic development, however, there are some concerns detailed below:

Answer: We thank the reviewer for recognizing the interesting aspects of our findings relative to the potential new functions of NF- κ B in DMD pathology.

1) *Title is misleading, this is more accurately “...through repression of calcium homeostatic genes ...” as opposed to “global epigenetic repression” as they show no evidence for any true epigenetic regulation (DNA methylation or polycomb/trithorax involvement). Similarly, there is no data presented on chromatin conformation, just chromatin content and only changes in active chromatin marks, no attempts to assay repressive marks (e.g. DNA methylation or H3K9me3 or H3K27me). Histone acetylation is typically associated with active chromatin but is often just an interaction face for transcriptional activators. Do DNaseI hypersensitive sites ever change in absence of NF- κ B? 3C interactions?*

Answer: We agree that it is important to be accurate with the epigenetics language used in our study, and thus have consulted with Dr. Mark Parthun on our campus who is an epigenetics expert and studies the regulation of chromatin in yeast and mammalian cells [<https://medicine.osu.edu/bcpharm/directory/faculty-directory/mark-r-parthun/pages/index.aspx>]. On his recommendation, we have removed the word “epigenetic” from our title and throughout the text and replaced it with “chromatin” or “chromatin-mediated”. However, we do believe that regulation at the chromatin level is occurring in our study. To demonstrate this, we have now added new ChIP data for both H3K9me3 and H3K27me3, which are accepted chromatin marks of repressed

transcription. These data in revised Fig. 3F and 3G show that in conjunction with increased H3K27ac, we see a depletion of H3K9me3 and H3K27me3 marks in p65 null MEFs compared to p65 wt MEFs.

2) *Line 196: Should be noted that p65(RelA) is reported to directly interact with HDAC1 and also function as a repressor in certain contexts so this is not completely unexpected.*

Answer: We apologize to the reviewer, as it was not our intention to ignore this finding, which actually points to the novelty of our paper. Previously, HDAC1 was shown to bind p65 and inactivate it through deacetylation. This is completely different from the mechanism we are showing where p65 is cooperating with HDAC1 to deacetylate chromatin. We have added a paragraph in the discussion section on potential use of HDAC inhibitors. In this section we discuss that p65 has been shown previously to directly interact with HDAC1 (see revised page 20).

3) *Line 201: Why a surprise? Typically, you are activating gene expression (e.g. pro-inflammatory genes) with NF-κB activation but also affecting a major gene regulatory network; p65 can be a repressor interacting with HDAC1, thus they are removing a known repressor. Also, p50 homodimers are repressive, what happens to their distribution and formation; how does removal of p65 affect the rest of the NF-κB dimer and heterodimer formations and targets?*

Answer: Please refer to our response to your point 2. In summary, previous reports of HDAC1 being a repressor with p65 is due to p65 being deacetylated by HDAC1, so it no longer activates its direct targets. There are no reports to our knowledge of p65 and HDAC1 repressing genes that are not bona fide targets for NF-κB. Regarding p50, we did not pursue p50 because *Slc8a1* expression did not increase in p50 null MEFs (Supplementary Fig. 2F) as expected if p50 homodimers were acting as a repressor complex.

4) *Have an issue that NF-κB (p65/p50 heterodimer) is a global repressor in mdx; don't think they have made the case. It would be very helpful if expression was put into the context of WT hearts.*

Answer: We thank the reviewer for this suggestion, as adding wt heart expression data has strengthened our paper and clarified the role of NF-κB in dystrophic hearts. We also agree now that calling NF-κB a global repressor in *mdx* hearts is not representative of what we are observing and have removed this language from the manuscript. As seen in revised Figs. 2D and 4F, all four of our target genes were expressed in wt hearts, indicating that none of these genes are epigenetically silenced in cardiac muscle. Although we observe decreases in H3K27ac when NF-κB is activated in *mdx* hearts, this loss of enrichment does not translate to a general downregulation of *Slc8a1* or the other calcium handling genes, compared to wt hearts. However, what is clear is that inhibiting NF-κB in *mdx* hearts enhances calcium gene expression concurrent with the restoration of H3K27ac and the concomitant decrease in CTCF, HDAC1, and Sin3a chromatin association. This suggests, as we describe in the Discussion section of our revised manuscript (see revised page 20), that in a dystrophic condition, the heart is trying to compensate for its functional deficiency by attempting to stimulate the transcription of calcium handling genes. However, this compensatory reaction is negated through the activation of NF-κB and association with CTCF, HDAC1, and Sin3a, which reduces H3K27ac and renders chromatin less active. It is only when we inhibit NF-κB that we see a more permissive chromatin state on calcium homeostatic genes, which in turn increases their gene expression and returns cardiac function back to wt conditions. In light of our new data, we have also modified the title of our manuscript accordingly.

5) *Supp Fig 3C, only a moderate 2-fold enrichment for H3K27Ac at Slc8a1 regulatory region? Is this just loss of repression and not really activation?*

Answer: Yes, we believe that this is loss of repression. We agree that the H3K27ac enrichment comparing wt and null p65 MEFs on *Slc8a1* is less than one might expect for direct gene activation. As discussed in the

previous point, *Slc8a1* and other calcium handling genes are not silenced in wt or *mdx* hearts, but are held in tight regulation by NF- κ B through chromatin modification as observed in *mdx* hearts, while ablation of NF- κ B increases the expression of the calcium homeostatic genes moderately. These data are presented in revised Figs. 2D and 4F.

6) Are any of these genes differentially methylated in WT vs mdx or +/- p65? It would be somewhat surprising for chronically (?) silenced genes, such as these appear in mdx, with CpG islands to not be methylated, but this is not addressed at any level. The general lack of information on the repression mechanism (other than HDAC1/2 which is in many, many chromatin complexes) for a manuscript with "repression" in the title is frustrating.

Answer: We apologize that this was unclear in our paper. No, none of the genes were chronically silenced either in wt or *mdx* hearts. As explained above, we have modified our position that activation of NF- κ B in dystrophic hearts is acting as a global repressor of calcium handling genes, and thus have removed the word "repression" from the title. Additionally, we have added text to the results on revised page 13 where we discuss the CpG island in *Slc8a1*, indicating that in addition to regulating genes through DNA methylation, CpG islands also influence chromatin conformation. This statement further assists in clarifying the confusion that we were not referring to DNA silencing rather than chromatin remodeling in this manuscript.

7) Authors state that they investigated several repressor proteins and only SIN3A and HDAC1 were present. Should list which ones and include e data as supplement as it will strengthen the specificity of the assays. Overall, I am cautious with data that shoes similar changes globally for a very specific phenotype.

Answer: We have removed this statement and instead focused on SIN3A and HDAC1.

8) Comparisons throughout are all in mdx with or without p65, however, it is important to know the comparison with WT hearts. Gene expression levels change and histone marks change but are they back to normal healthy levels and patterns?

Answer: We recognize this is an important point, and thus have included new data to address it in our manuscript. Wt gene expression data has now been added to revised Figs 2D and 4F, which shows that all 4 genes are expressed in wt hearts. We have additionally added a H3K27ac ChIP from wt heart tissue for *Slc8a1* (Fig 3J). These data show that H3K27ac returned to near wt levels in the absence of NF- κ B activation.

9) It is surprising that DNA methylation was not really investigated since the Slc8a1 regulatory region is a CpG island and they over riding claim is an epigenetic mechanism. Regardless, the data supports typical transcription factor-mediated repression/activation at these genes and not true epigenetic regulation. The aza-C experiment was weak, particulaly since AzaC can in instances activate gene expression without affecting DNA methylation so it is not really clear the control for demethylation even worked.

Answer: As discussed above, we have refrained from claiming that effects were due to epigenetic regulation, which is why throughout our manuscript, and in the title, we use the terms "chromatin-mediated" or simply "chromatin remodeling", which we believe better reflects the chromatin remodeling events as opposed to a more significant gene silencing mediated by methylation. This is supported by wt gene expression data of our target genes in hearts (Figs 2D and 4F), results from ChIPs on wt hearts for H3K27ac on *Slc8a1* (Fig 3J), results from ChIP assays for the chromatin repressive marks H3K9me3 (Fig 3F) and H3K27me3 (Fig 3G) on MEFs.

REVIEWERS' COMMENTS:

Reviewer #1 (Remarks to the Author):

The authors have been responsive to my comments (as well as to other reviewers comments) and the manuscript deserves to be published, as it contains information of great importance from a biological standpoint as well as in the field of DMD therapeutics

Reviewer #2 (Remarks to the Author):

(No further comments from this reviewer)

Reviewer #3 (Remarks to the Author):

In the revised article entitled "NF- κ B promotes cardiac dysfunction through global chromatin-mediated remodeling of calcium homeostatic genes in a model of Duchenne muscular dystrophy" by Peterson et. al., the authors have included WT mice, performed histology in 1 year old hearts to match the functional data and included 2 Tables with more functional parameters. However, they still omitted the HRTDeltaIKK2beta mice in all their studies. This is a major concern and it is critical in order to properly interpret all data reported in this paper. More specifically:

1. The comparison between the double mutant (mdxHRTDIKKb) and the individual mutants (mdxIKKbf/f and HRTDeltaIKK2beta) alone will definitely demonstrate that any cardiac measurement or chromatin modification is not due to dystrophin mutation alone or due to cardiac deletion of IKKbeta per se, but rather due to the combined dystrophin deficiency with cardiac-specific IKKbeta deletion. Otherwise, the authors cannot exclude that every phenotype or any other significant difference reported in this paper (including all Figures and Tables) is not the result of the HRTDeltaIKK2beta alone.

2. The authors suggest that in "a dystrophic condition" (mdx alone), "the heart is trying to compensate for its functional deficiency by attempting to stimulate the transcription of calcium handling genes". The suggestion is that the "compensatory reaction is negated through the activation of NF- κ B and association with CTCF, HDAC1, and Sin3a, which reduces H3K27ac and renders chromatin less active". They claim that "it is only when NF- κ B is inhibited that a more permissive chromatin state on calcium homeostatic genes occurs, which in turn increases their gene expression and returns cardiac function back to WT conditions". This argument cannot be supported without the analysis of the HRTDeltaIKK2beta mice.

3. In Figures 1 J and K, the NBD (NEMO Binding Domain Peptide) on mdx mice (mdx-NBD) behave similarly with mdxHRTDIKKbeta mice. However, discussing (see lines 468-470 of the revised manuscript) the cardiac phenotyping differences of the mdxHRTDeltaIKKbeta mice with the phenotype of cardiac-specific deletion of NEMO mice (Ref 50), the authors propose that "differences in deletion of IKKbeta and NEMO or potentially due to difference in phenotype when NF- κ B signaling is interrupted in a disease state (dystrophy) rather than in a normal condition". Without including the HRTDeltaIKK2beta mice, this is a weak argument, especially because they use a peptide that acts on the NEMO subunit to block the IKK complex.

Minor Comments:

1. Evidence of the reduced level of IKKbeta deletion by QRT-PCR, Western blot or both is necessary.
2. Statement on page 12, line 248 "global repressive effect " is weak. There actually were not many genes altered.
3. In the microarray, there is no mention of cut-off of 1.5 fold difference is used. Typical is a cut-off of 2, which helps with natural biological variation.
4. It is unclear why staining in Figure 1C is yellow and not brown. In addition, why the p65 staining in Supplementary Figure 1D and E is perinuclear and not nuclear?

Reviewer #4 (Remarks to the Author):

This is a revision submission investigating the role of NF- κ B mediated chromatin changes in the DMD heart. The main findings are that in DMD, NF- κ B functions pathologically as a repressor of calcium homeostasis genes through chromatin remodeling on specific enhancer elements and inhibition of NF- κ B activity removes this repression, increases calcium homeostatic gene expression, and rescues the cardiac dysfunction.

In this revision, the authors have addresses all of my initial concerns. The extensive additional data and revised textual changes have greatly improved the paper and strongly support their claims and has increased the impact on the field. The statistics are appropriate and adequate.

Specific Responses to Reviewer #3

1). *The comparison between the double mutant (mdxHRTDIKKb) and the individual mutants (mdxIKKbf/f and HRTDeltaIKK2beta) alone will definitely demonstrate that any cardiac measurement or chromatin modification is not due to dystrophin mutation alone or due to cardiac deletion of IKKbeta per se, but rather due to the combined dystrophin deficiency with cardiac specific IKKbeta deletion. Otherwise, the authors cannot exclude that every phenotype or any other significant difference reported in this paper (including all Figures and Tables) is not the result of the HRTDeltaIKK2beta alone.*

Answer: The reviewer makes an important point, which we had addressed in our previous pt-by-pt responses by stating that NF- κ B might control the homeostatic function of cardiac muscle through a similar mechanism to what we described in our current study in *mdx* hearts. To make this more transparent to the reader, we have now included text in our Discussion section on revised page 17-18, which reads, *"It will be interesting in future studies to address whether the ability of NF- κ B to regulate H3K27ac and restrict chromatin confirmation occurs under non-pathological conditions in normal cardiomyocytes, or as shown with C2C12 and MEFs, is relevant in other non-cardiac tissues"*.

However, we would like to stress that the goal of our study was to determine how NF- κ B contributes to cardiomyopathy in dystrophic mice, given that this has not been addressed before, and not necessarily to determine its physiological function in the heart. This is the reason we did not include a separate cohort of HRTDeltaIKK2beta. It is also the same reason why in past studies we did not include skeletal muscle DeltaIKK2beta mice when trying to understand how NF- κ B contributes to the pathology of skeletal muscles in *mdx* mice (Archayya et al., 2007, JCI) or satellite cell DeltaIKK2beta to understand how NF- κ B contributes to the pathology of skeletal muscles in cancer cachexia (He et al., 2013, JCI).

We believe our current findings showing that NF- κ B contributes to cardiomyopathy of *mdx* hearts is a novel finding. Similarly, showing that NF- κ B functions as a repressor of calcium genes, and this function is mediated by regulating H3K27ac through the recruitment of co-repressor proteins, is also novel, which at least for the DMD field should be sufficient to be reported as a stand alone study. We also want to reiterate our Discussion point that, to the best of our knowledge, NF- κ B represents the first signaling pathway capable of promoting pathology in both dystrophic cardiac and skeletal muscle, thus providing a strong rationale for targeting this pathway as a therapeutic for DMD. We agree with the reviewer that we cannot discount the possibility that NF- κ B regulation of H3K27ac on calcium genes might occur independently of dystrophin deficiency, but we do feel our findings nevertheless conceptually advance our understanding of the role of NF- κ B signaling in *mdx* cardiomyopathy.

2). *The authors suggest that in "a dystrophic condition" (mdx alone), "the heart is trying to compensate for its functional deficiency by attempting to stimulate the transcription of calcium handling genes". The suggestion is that the "compensatory reaction is negated through the activation of NF-kB and association with CTCF, HDAC1, and Sin3a, which reduces H3K27ac and renders chromatin less active". They claim that "it is only when NF-kB is inhibited that a more permissive chromatin state on calcium homeostatic genes occurs, which in turn increases their gene expression and returns cardiac function back to WT conditions". This argument cannot be supported without the analysis of the HRTDeltaIKK2beta mice.*

Answer: Similar to our response in pt #1 above, we agree that a similar scenario to what we describe in dystrophic hearts might also occur under homeostatic conditions. However, the point we were making with how the heart might compensate in a dystrophic condition was only meant to provide a potential explanation for why we were unable to see decreases in the expression of calcium genes in *mdx* hearts.

This explanation was only meant as speculation, which is why it appeared in the Discussion section. To test this idea properly will also require a separate study, at which time the HRTDeltaIKK2beta mice can be included as a control cohort and tested at different times of embryonic and post-natal development.

3. In Figures 1 J and K, the NBD (NEMO Binding Domain Peptide) on mdx mice (mdx-NBD) behave similarly with mdxHRTDIKKbeta mice. However, discussing (see lines 468-470 of the revised manuscript) the cardiac phenotyping differences of the mdxHRTDeltaIKKbeta mice with the phenotype of cardiac-specific deletion of NEMO mice (Ref 50), the authors propose that "differences in deletion of IKKbeta and NEMO or potentially due to difference in phenotype when NF-kB signaling is interrupted in a disease state (dystrophy) rather than in a normal condition". Without including the HRTDeltaIKK2beta mice, this is a weak argument, especially because they use a peptide that acts on the NEMO subunit to block the IKK complex.

Answer: The intent of bringing up this point in the Discussion section was to make clear to the reader that significant differences exist in the way NF- κ B signaling has been found to function in normal and diseased hearts, and that the variability in these findings might lie in whether one is studying a physiological or disease condition. Although the reviewer makes a valid point regarding the use of HRTDeltaIKK2beta to compare directly with HRTDeltaNEMO hearts, it is difficult to understand that if deletion of IKKbeta in normal hearts phenocopied the pathology with NEMO, why would the addition of a disease allele such as *mdx* improve, not worsen, heart function as we have seen? Thus, we believe at this juncture, it is important to point out that not every inhibition of the NF-kB pathway ever attempted results in improved cardiac function.

Minor Comments:

1. Evidence of the reduced level of IKKbeta deletion by qRT-PCR, Western blot or both is necessary.

Answer: We have revised the manuscript to include qRT-PCR data that reflects lower IKKbeta expression in the hearts of our knock out mice (Supplementary Figure 1B). Please note that there will still be some IKKbeta that remains because the qRT-PCR reactions were performed on whole hearts, which contain unaltered fibroblasts and endothelial cells, in addition to cardiomyocytes depleted of IKKbeta.

2. Statement on page 12, line 248 "global repressive effect " is weak. There actually were not many genes altered.

Answer: This statement has now been modified to more specifically reflect the overall changes in gene expression altered by loss of IKKbeta in the hearts of dystrophic mice. However, although our identified changes in gene expression were modest, we maintain that NF- κ B signaling in dystrophic hearts has a global repressive effect on chromatin structure since approximately 10,000 additional H3K27ac peaks were identified in the cardiomyocytes lacking IKKbeta (see Fig. 4A).

3. In the microarray, there is no mention of cut-off of 1.5 fold difference is used. Typical is a cut-off of 2, which helps with natural biological variation.

Answer: The cut-off values that we used for the microarray analysis are listed in the text (page 9) and legends of Fig. 2. While we acknowledge that a cut-off of 2 is common, we elected to use a 1.5 cut-off given that whole hearts are a mixed population of cells, and using a higher cut-off would miss many biologically relevant genes specific to cardiomyocytes.

4. It is unclear why staining in Figure 1C is yellow and not brown. In addition, why the p65 staining in Supplementary Figure 1D and E is perinuclear and not nuclear?

Answer: The staining is DAB and to us appears brown. We suspect the yellowish hue is due to some background staining from the cardiomyocyte cytoplasmic compartment. As for the p65 staining, we agree that the signal looks perinuclear, which is similar to the staining we obtained for p65 in *mdx* dystrophic skeletal muscle (Acharrya et al., 2007, JCI). This phenomenon could be a caveat of *mdx* nuclei since this p65 serine 536 antibody is commonly used by other labs on non *mdx* tissues and similar perinuclear staining has not been reported.